# Causal Inference with Attention: on duality between optimal balancing and attention

## Abstract

Foundation models have brought changes to the landscape of machine learning, demonstrating sparks of human-level intelligence across a diverse array of tasks. However, a gap persists in complex tasks such as causal inference, primarily due to challenges associated with intricate reasoning steps and high numerical precision requirements. In this work, we take a first step towards building causally-aware foundation models for treatment effect estimations. We propose a novel, theoretically sound method called Causal Inference with Attention (CInA), which utilizes multiple unlabeled datasets to perform self-supervised causal learning, and subsequently enables zero-shot causal inference on unseen tasks with new data. This is based on our theoretical results that demonstrate the primal-dual connection between optimal covariate balancing and self-attention, facilitating zero-shot causal inference through the final layer of a trained transformer-type architecture. We demonstrate empirically that CInA effectively generalizes to out-of-distribution datasets and various real-world datasets, matching or even surpassing traditional per-dataset methodologies. These results provide compelling evidences that our method has the potential to serve as a fundamental building block for the development of causal foundation models.

## 1 Introduction

Recent advances in artificial intelligence have created a paradigm shift in which models are trained on large amounts of data and can be adapted to different tasks, dubbed *foundation models* (Bommasani et al., 2021). These models, which often employ self-supervision, can extract valuable knowledge from various types of data, including natural language (Devlin et al., 2018; Brown et al., 2020), images (Radford et al., 2021), and biological sequencing counts (Theodoris et al., 2023). This acquired knowledge allows the model to generalize when asked to perform tasks in novel scenarios. With vast amounts of data becoming increasingly available from diverse sources, such models are of interest to leverage information that can be learned in order to build more intelligent systems (Bubeck et al., 2023).

A critical aspect of intelligent systems is the ability to reason about cause-and-effect relationships, which is vital to making informed decisions across various domains, including healthcare, economics, and statistics (Harrison & March, 1984; Kube et al., 2019; Geffner et al., 2022; Zhang et al., 2023c). There have been significant debates regarding whether current foundation models acquire the ability to reason about causality (Kıcıman et al., 2023; Zečević et al., 2023). However, it was observed that existing foundation models have difficulties with causal tasks that involve intricate reasoning or high numerical precision (Bubeck et al., 2023; Mahowald et al., 2023; Wolfram, 2023; Zečević et al., 2023; Jin et al., 2023), such as treatment effect estimations. Furthermore, performance may decline when tested on datasets that were not part of the training set (Feder et al., 2022). Motivated by this shortcoming, it is crucial to build *causally-aware foundation models (see Appendix A a definition)* capable of extracting causal information and performing causal inference at scale, harnessing the vast amounts of data available from diverse sources.

However, creating a suitable self-supervised learning paradigm for causal foundation models with theoretical guarantees remains an open question. Unlike existing foundational models for natural language and vision (e.g., Devlin et al. (2018); Radford et al. (2021)), causal foundation models generally lacks clearly defined supervised signals since most available machine learning datasets

only contain observational data without intervention, rendering key causal quantities, such as treatment effects, unknown. On top of this, common datasets used in causality community contain complex relationships between variables that might be heterogeneous across dataset sources. These less-structured heterogeneous relationships make it harder for the model to capture compared to linguistic or perceptual patterns.

**Contributions.** In this paper, we take a *first step* towards building causal foundation models, focusing on estimating treatment effects with greater generalizability. One of our primary contributions is a theoretically sound method, dubbed **C**ausal **In**ference with **A**ttention (CInA), that leverages multiple unlabeled observational datasets to learn how to estimate treatment effects on various tasks, and then perform zero-shot causal inference on unseen tasks with new data.

- We theoretically establish the equivalence between optimal covariate balancing and (regularized) self-attention through a primal-dual argument. We prove that with an appropriate self-supervised loss, a trained self-attention is guaranteed to find the optimal balancing weights for any given dataset under certain regularity conditions. This serves as the theoretical foundation that enables zero-shot causal inference on unseen data.

- Based on our theoretical results, we propose a gradient-based, transformer-type practical algorithm for zero-shot causal inference. In particular, this model uses covariate balancing as self-supervised tasks. Once trained on multiple data sources, it performs zero-shot causal inference by simply extracting the key-value tensors from the last layer of the model during a forward pass on new data. This stands in contrast to traditional per-dataset causal inference, which needs to re-fit and re-optimize on new data.

- Empirically, we verify the correctness of our theory and demonstrate the effectiveness of our algorithm on both synthetic and real-world datasets. Importantly, in the context of zero-shot causal inference on unseen datasets, we observed competitive and in-certain-cases better performance to traditional per-dataset causal inference approaches, while achieving substantial reductions in inference time.

These results show evidence that the proposed method can serve as a *fundamental building block* in the development of causally-aware foundation models.

**Organization.** In Section 2, we discuss related works. In Section 3, we state our theoretical results and provide the derivation of our algorithm, which serves as a proof sketch. We use these results to derive our methods for zero-shot causal inference in Section 4. In Section 5, we perform empirical studies of our proposed algorithms on both synthetic and real-world datasets.

## 2    RELATED WORKS

**Causal Inference via Optimal Balancing.** Our work concerns problems in causal inference, assuming that we are provided with either the causal structure (Pearl, 2009) or certain independence conditions between variables that imply structural relationships (Imbens & Rubin, 2015). In particular, we focus on estimation problems, e.g., estimating average treatment effect (ATE) and policy evaluation. See Section 3.1 for a detailed problem formulation. Under certain assumptions, one of the most common methods is to use weighted (e.g., Li et al. (2018)) or doubly robust estimators (e.g., Dudík et al. (2011)). Numerous weighted estimators have been proposed to optimize covariate balance (e.g., Hainmueller (2012); Imai & Ratkovic (2014)). Our work extends this line of research by introducing an optimal balancing approach that relies on training a transformer-type model, which is the main architecture used by existing foundation models (Bommasani et al., 2021). We discuss related neural estimation method in Appendix B. It is worth noting that we also differ from prior work by considering multiple datasets simultaneously, where we show that our proposed method can be generalized to produce estimand on a new dataset in a zero-shot manner.

**Causal Reasoning with Large Language Models (LLMs).** A prominent example of foundation models are LLMs (Brown et al., 2020; OpenAI, 2023). Due to their remarkable performance across various tasks, prior works have explored and exploited their capabilities in addressing causal inquiries. For example, (Zhang et al., 2023a) assessed the ability of LLMs for three types of causal questions: identifying causal relationships using existing domain knowledge, discovering new knowledge from data, and estimating quantitative treatment effects. They found that LLMs perform well on the first question but are not yet to provide satisfactory answers for the others. Similar

limitations with formal reasoning have also been noted in (Bubeck et al., 2023; Mahowald et al., 2023; Wolfram, 2023). When probing LLMs, Li et al. (2022); Park et al. (2023) found evidences of emergent representations that are helpful for causal predictions. However, it was observed that for causal discovery, LLMs are not yet stable (Kıcıman et al., 2023) and might produce different answers to the same question in two separate queries (Tu et al., 2023). To enhance LLMs for causal tasks, Ban et al. (2023) proposed to integrate LLM outputs with constraint-based methods.

In this paper, we take a different path towards causally-aware foundation models; namely, we explore the fundamentals of constructing these models from scratch to address questions on a larger scale and with greater generalizability than current statistical tools. It is important to note that, apart from utilizing the attention architecture, this work has no further connection with LLMs.

## 3 ESTABLISHING DUALITY BETWEEN CAUSALITY AND ATTENTION

We present our main theoretical result on the primal-dual connection between covariate balancing and self-attention, which enables us to estimate treatment effects via transformer-type architectures. In particular, in Section 3.1, we describe the adversarial optimal balancing formulation of causality and show how optimal balancing can be viewed as a specific dual support vector machine (SVM) problem. Then, in Section 3.2, we establish the equivalence between the SVM expansion and self-attention. Detailed derivations of this section can be found in Appendix C.

### 3.1 ADVERSARIAL COVARIATE BALANCING AS DUAL SVM

To illustrate our approach, we focus on the task of average treatment effect estimation. In Appendix F, we extend our method to other estimands, such as individual treatment effect and policy evaluation. Consider a dataset of $N$ units $\mathbb{D} = \{(\boldsymbol{X}_i, T_i, Y_i)\}_{i \in [N]}$, where $\boldsymbol{X}_i$ is the observed covariates, $T_i$ is the observed treatment, and $Y_i$ is the observed outcome. Suppose $T_i \in \{0, 1\}$ for now; Appendix E generalizes these results for non-binary treatments. Let $Y_i(t)$ be the potential outcome of assigning treatment $T_i = t$. The sample average treatment effect is defined as $\tau_{SATE} = \frac{1}{N} \sum_{i=1}^{N} (Y_i(1) - Y_i(0))$.

Assume $Y_i = Y_i(T_i)$, i.e., consistency between observed and potential outcomes and non-interference between units (Rubin, 1990), and $Y_i(0), Y_i(1) \perp T_i \mid \boldsymbol{X}_i$, i.e., no latent confounders. We consider weighted estimators in the form of

$$\hat{\tau} = \sum_{i \in \mathbb{T}} \alpha_i Y_i(1) - \sum_{i \in \mathbb{C}} \alpha_i Y_i(0),$$

where $\mathbb{T} = \{i \in [N] : T_i = 1\}$ is the treated group and $\mathbb{C} = \{i \in [N] : T_i = 0\}$ is the control group. We force constraints on the weight by allowing $\boldsymbol{\alpha} \in \mathbb{A} = \{\boldsymbol{0} \preceq \boldsymbol{\alpha} \preceq \boldsymbol{1}, \ \sum_{i \in \mathbb{T}} \alpha_i = \sum_{i \in \mathbb{C}} \alpha_i = 1\}$. These constraints help with obtaining robust estimators. For example, $\sum_{i \in \mathbb{T}} \alpha_i = 1$ ensures that the bias remains unchanged if we add a constant to the outcome model of the treated, whereas $\sum_{i \in \mathbb{C}} \alpha_i = 1$ further ensures that the bias remains unchanged if we add the same constant to the outcome model of the control.

A good estimator should minimize the absolute value of the conditional bias that can be written as

$$\mathbb{E}\left(\hat{\tau} - \tau_{SATE} \mid \{\boldsymbol{X}_i, T_i\}_{i=1}^{N}\right) = \sum_{i=1}^{N} (\alpha_i T_i - \frac{1}{N}) \mathbb{E}\left(Y_i(1) - Y_i(0) \mid \boldsymbol{X}_i\right) + \sum_{i=1}^{N} \alpha_i W_i \mathbb{E}\left(Y_i(0) \mid \boldsymbol{X}_i\right),$$

where we denote $W_i = 1$ if $i \in \mathbb{T}$ and $W_i = -1$ if $i \in \mathbb{C}$. As the outcome models are typically unknown, we follow previous works (Tarr & Imai, 2021; Kallus, 2020b) by minimizing an upper bound on the square of the second term.[1] Namely, assuming the outcome model $\mathbb{E}(Y_i(0) \mid \boldsymbol{X}_i)$ belongs to a hypothesis class $\mathcal{F}$, we solve for $\min_{\boldsymbol{\alpha} \in \mathbb{A}} \sup_{f \in \mathcal{F}} \left(\sum_{i=1}^{N} \alpha_i W_i f(\boldsymbol{X}_i)\right)^2$. To simplify this, consider $\mathcal{F}$ being an unit-ball reproducing kernel Hilbert space (RKHS) defined by some feature map $\phi$. Then the supremum can be computed in closed form, which reduces the optimization problem to

$$\min_{\boldsymbol{\alpha} \in \mathbb{A}} \boldsymbol{\alpha}^\top \boldsymbol{K}_\phi \boldsymbol{\alpha}, \tag{1}$$

---

[1] In Appendix D, we show how our method can generalize to alternative balancing objectives, e.g., the square of both terms in the conditional bias and the conditional mean square error.

where $[\boldsymbol{K}_\phi]_{ij} = W_i W_j \langle \phi(\boldsymbol{X}_i), \phi(\boldsymbol{X}_j) \rangle$. Here $\langle \cdot, \cdot \rangle$ denotes the inner product of the Hilbert space to which $\phi$ projects. This is equivalent to solving the following dual SVM problem for some $\lambda \geq 0$ (Theorem 1 in Tarr & Imai (2021)),

$$\min_{\boldsymbol{\alpha}} \quad \boldsymbol{\alpha}^\top \boldsymbol{K}_\phi \boldsymbol{\alpha} - 2\lambda \cdot \mathbf{1}^\top \boldsymbol{\alpha},$$
$$s.t. \quad \boldsymbol{W}^\top \boldsymbol{\alpha} = 0, \quad \mathbf{0} \preceq \boldsymbol{\alpha} \preceq \mathbf{1}. \tag{2}$$

In other words, the optimal solution $\boldsymbol{\alpha}^*$ to Eq. (2) solves Eq. (1). Thus we can obtain the optimal balancing weight by solving the dual SVM. For the choice of the RKHS, we will see in the next section that the feature function $\phi$ is also learned from data.

## 3.2 SELF-ATTENTION AS SUPPORT VECTOR EXPANSION

**SVM to Self-attention.** The dual SVM problem for covariate balancing (Eq. (2)) has the following primal form:

$$\min_{\boldsymbol{\beta}, \beta_0, \boldsymbol{\xi}} \quad \frac{\lambda}{2} \|\boldsymbol{\beta}\|^2 + \sum_{i=1}^{N} \xi_i,$$
$$s.t. \quad W_i \left( \langle \boldsymbol{\beta}, \phi(\boldsymbol{X}_i) \rangle + \beta_0 \right) \geq 1 - \xi_i, \quad \xi_i \geq 0, \quad \forall i \in [N]. \tag{3}$$

Intuitively, this optimization problem aims to classify the treatment assignment $W_i$ using a linear transformation of the feature vector $\phi(\boldsymbol{X}_i)$.

We can connect the primal solution to the dual coeffcients $\boldsymbol{\alpha}^*$ by the Karush-Kuhn-Tucker (KKT) condition (Boyd & Vandenberghe, 2004). The optimal $\boldsymbol{\beta}^*$ that solves Eq. (3) should satisfy $\lambda \boldsymbol{\beta}^* = \sum_{j=1}^{N} \alpha_j^* W_j \phi(\boldsymbol{X}_j)$. Thus if $\lambda > 0$, the optimal classifer will have the following support vector expansion

$$\langle \boldsymbol{\beta}^*, \phi(\boldsymbol{X}_i) \rangle = \sum_{j=1}^{N} (\alpha_j^* W_j / \lambda) \cdot \langle \phi(\boldsymbol{X}_j), \phi(\boldsymbol{X}_i) \rangle. \tag{4}$$

Note that we drop the constant intercept for simplicity. Next we show how Eq. (4) relates to self-attention.

Consider input sequence as $\boldsymbol{X} = [\boldsymbol{X}_1, ..., \boldsymbol{X}_N]^\top \in \mathbb{R}^{N \times D_X}$. We use a self-attention layer to attend to units in a dataset instead of words in a sentence (Vaswani et al., 2017), as illustrated in Figure 1. This can be expressed as

$$\text{softmax}\left(\boldsymbol{Q}\boldsymbol{K}^\top / \sqrt{D}\right)\boldsymbol{V},$$

where $\boldsymbol{Q} = [\boldsymbol{q}_1, ..., \boldsymbol{q}_N]^\top \in \mathbb{R}^{N \times D}$, $\boldsymbol{K} = [\boldsymbol{k}_1, ..., \boldsymbol{k}_N]^\top \in \mathbb{R}^{N \times D}$, and $\boldsymbol{V} = [v_1, ..., v_N]^\top \in \mathbb{R}^{N \times 1}$. Here we consider output as a sequence of scalars; in general, $\boldsymbol{V}$ can be a sequence of vectors. The query and key matrices $\boldsymbol{Q}, \boldsymbol{K}$ can be $\boldsymbol{X}$ itself or outputs of several neural network layers on $\boldsymbol{X}$.

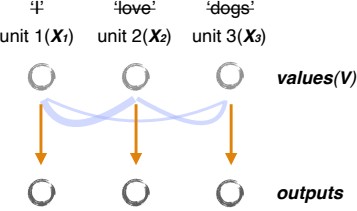

Figure 1: Attending to units instead of words. Values correspond to covariate balancing weights.

Note that the softmax operation is with respect to per column of $\boldsymbol{QK}^\top/\sqrt{D}$, i.e., the $i$-th output is

$$\sum_{j=1}^{N} \frac{\exp\left(\boldsymbol{q}_i^\top \boldsymbol{k}_j / \sqrt{D}\right)}{\sum_{j'=1}^{N} \exp\left(\boldsymbol{q}_i^\top \boldsymbol{k}_{j'} / \sqrt{D}\right)} v_j. \tag{5}$$

Following Nguyen et al. (2022), if we set $\boldsymbol{Q} = \boldsymbol{K}$, then there exists a feature map (exact form given in Appendix C) such that for any $i, j \in [N]$, there is $\langle \phi(\boldsymbol{X}_j), \phi(\boldsymbol{X}_i) \rangle = \exp\left(\boldsymbol{k}_i^\top \boldsymbol{k}_j / \sqrt{D}\right)$. Let $h(\boldsymbol{X}_i) = \sum_{j'=1}^{N} \exp(\boldsymbol{k}_i^\top \boldsymbol{k}_{j'} / \sqrt{D})$. We can rewrite the $i$-th output of attention layer in Eq. (5) as

$$\sum_{j=1}^{N} \frac{v_j}{h(\boldsymbol{X}_j)} \langle \phi(\boldsymbol{X}_j), \phi(\boldsymbol{X}_i) \rangle. \tag{6}$$

This recovers the support vector expansion in Eq. (4) by setting $\lambda v_j / h(\boldsymbol{X}_j) = \alpha_j^* W_j$. This shows that at optimum, the SVM classifier takes the form of self-attention.

**Self-attention to SVM.** Conversely, under mild regularities, we can also read off the optimal balancing weight $\alpha_j^*$ from $\lambda v_j / h(\boldsymbol{X}_j) W_j$ if the attention layer is globally optimized with an appropriate loss function. In particular, with a penalized hinge loss, the learned optimal self-attention will solve the primal SVM problem in Eq. (3). Then by the primal-dual relationship, we can equate Eq. (6) with Eq. (4). This establishes the duality between self-attention and the optimal balancing weights $\boldsymbol{\alpha}^*$, which is summarized in Theorem 1. The details of Algorithm 1 can be found in Section 4.1.

**Theorem 1** (Duality between covariate balancing and self-attention). *Under mild regularities on $\boldsymbol{X}$, learning a self-attention via gradient-based Algorithm 1 recovers the optimal covariate balancing weight at the global minimum of the penalized hinge loss in Eq. (7).*

## 4 PRACTICAL ALGORITHMS TOWARDS CAUSAL FOUNDATION MODELS

In this section, we show how our theoretical results can lead to a gradient-based, transformer-type algorithm for zero-shot optimal covariate balancing. Specifically, in Section 4.1, we introduce a gradient-based solution for the traditional single-dataset setting. We then show how it can be extended to enable zero-shot inference on unseen datasets through amortization in Section 4.2. Details of the model architecture and preprocessing steps are provided in Appendix H.

### 4.1 GRADIENT-BASED OPTIMAL BALANCING VIA SELF-ATTENTION

Comparing Eq. (6) and Eq. (4), we seek a training procedure such that $\sum_{j=1}^{N} \frac{v_j}{h(\boldsymbol{X}_j)} \phi(\boldsymbol{X}_j)$ recovers the optimal $\boldsymbol{\beta}^*$ that solves primal SVM in Eq. (3). Note that Eq. (3) corresponds to a constrained optimization problem that is unsuitable for gradient descent methods. However, it is equivalent to an unconstrained optimization problem by minimizing the penalized hinge loss (Hastie et al., 2009) $\frac{\lambda}{2} \|\boldsymbol{\beta}\|^2 + \sum_{i=1}^{N} \left[ 1 - W_i \left( \langle \boldsymbol{\beta}, \phi(\boldsymbol{X}_i) \rangle + \beta_0 \right) \right]_+$. This motivates the use of the following loss function:

$$\mathcal{L}_{\boldsymbol{\theta}}(\mathbb{D}) = \frac{\lambda}{2} \left\| \sum_{j=1}^{N} \frac{v_j}{h(\boldsymbol{X}_j)} \phi(\boldsymbol{X}_j) \right\|^2 + \left[ \mathbf{1} - \boldsymbol{W} \left( \mathrm{softmax}(\boldsymbol{K}\boldsymbol{K}^\top / \sqrt{D}) \boldsymbol{V} + \beta_0 \right) \right]_+ . \tag{7}$$

Here we use $\boldsymbol{\theta}$ to subsume all the learned parameters, including $\boldsymbol{V}$ and parameters of the layers (if any) to obtain $\boldsymbol{K}$. We learn $\boldsymbol{\theta}$ via gradient descent on Eq. (7). Note that the penalization can be computed exactly by using the formula for inner products between features, i.e.,

$$\left\| \sum_{j=1}^{N} \frac{v_j}{h(\boldsymbol{X}_j)} \phi(\boldsymbol{X}_j) \right\|^2 = \sum_{i,j=1}^{N} \frac{v_i v_j \exp\left( \boldsymbol{k}_i \boldsymbol{k}_j^\top / \sqrt{D} \right)}{h(\boldsymbol{X}_i) h(\boldsymbol{X}_j)} .$$

Theorem 1 guarantees that under mild regularities, the optimal parameters lead to the optimal balancing weights in terms of the adversarial squared error. This adversarial squared error is computed using an unit-ball RKHS defined by $\phi$. The optimal balancing weights and ATEs can be obtained via

$$\alpha_j^* = \frac{\lambda v_j}{h(\boldsymbol{X}_j) W_j},$$
$$\hat{\tau} = (\boldsymbol{\alpha}^* \boldsymbol{W})^\top \boldsymbol{Y}.$$

---

**Algorithm 1** Causal Inference with Attention (CInA)

1: **Input**: Covariates $\boldsymbol{X}$ and treatments $\boldsymbol{W}$.
2: **Output**: Optimal balancing weight $\boldsymbol{\alpha}^*$.
3: Hyper-parameter: penalty weight $\lambda > 0$.
4: Parameters: $\boldsymbol{\theta}$ (including $\boldsymbol{V}$), step size $\eta$.
5: **while** not converged **do**
6:     Compute $\boldsymbol{K}$ using forward pass.
7:     Update $\boldsymbol{\theta} \leftarrow \boldsymbol{\theta} - \eta \nabla \mathcal{L}_{\boldsymbol{\theta}}$.
8: **return** $\lambda \cdot \boldsymbol{V} / h(\boldsymbol{X}) \boldsymbol{W}$.

---

Note that for this result to hold, arbitrary mappings can be used to obtain $\boldsymbol{k}_i$ from $\boldsymbol{X}_i$, thus allowing for the incorporation of flexible neural network architectures. We summarize our method in Algorithm 1, which is later referred to as *CInA* (or *Ours*).

### 4.2 ZERO-SHOT CAUSAL INFERENCE UNDER MULTI-DATASET SETTING

To enable zero-shot estimation of treatment effects, we consider multiple datasets denoted as $\mathbb{D}^{(m)} = \{(\boldsymbol{X}_i, T_i, Y_i)\}_{i \in [N_m]} = (\boldsymbol{X}^{(m)}, \boldsymbol{T}^{(m)}, \boldsymbol{Y}^{(m)})$ for $m \in [M]$. Each dataset $\mathbb{D}^{(m)}$ contains

$N_m$ units following the description in Section 3.1. We allow for datasets of different sizes, mimicking real-world data gathering practices, where a large consortium of datasets may exist. The setting encapsulates cases where individual datasets are created by distinct causal mechanisms; however, different units within a single dataset should be generated via the same causal model. This presents a new challenge, which requires the model to generalize to new datasets without supervision.

Algorithm 1 shows how one can read off the optimal weights $\boldsymbol{\alpha}^*$ from a trained model with attention as its last layer in a single dataset. Note that the value vector $\boldsymbol{V}$ is encoded as a set of parameters in this setting. On a new dataset $\mathbb{D}^{(*)} = (\boldsymbol{X}^{(*)}, \boldsymbol{T}^{(*)}, \boldsymbol{Y}^{(*)})$, the values of $\boldsymbol{X}^{(*)}$ and $\boldsymbol{W}^{(*)}$ are changed, and thus the optimal $\boldsymbol{V}^{(*)}$ that minimizes $\mathcal{L}_{\boldsymbol{\theta}}(\mathbb{D}^{(*)})$ should also differ from the encoded parameters. As indicated by the form of $\mathcal{L}_{\boldsymbol{\theta}}(\mathbb{D}^{(*)})$, the optimal $\boldsymbol{V}^{(*)}$ only depends on $\boldsymbol{X}^{(*)}$ through $\boldsymbol{K}^{(*)}$. Therefore we encode the value vector $\boldsymbol{V}$ as a transformation of $\boldsymbol{K}$ and $\boldsymbol{W}$. Denote the parameters of this transformation as $\phi$ and let $\boldsymbol{\theta}$ subsumes $\phi$. We learn $\phi$ by minimizing

$$\sum_{m \in [M]} \mathcal{L}_{\boldsymbol{\theta}}(\mathbb{D}^{(m)})$$

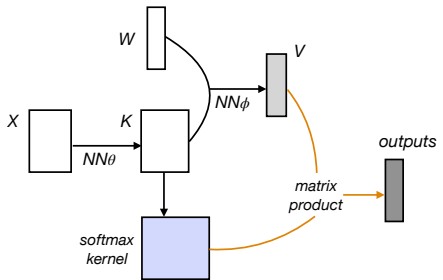

Figure 2: CInA(multi-dataset) forward pass.

on the training datasets in an end-to-end fashion. On a new dataset not seen during training, we can directly infer its optimal balancing weight $\boldsymbol{\alpha}^*$ via $\lambda \cdot \boldsymbol{V}^{(*)}/h(\boldsymbol{X}^{(*)})\boldsymbol{W}^{(*)}$, where $\boldsymbol{V}^{(*)}$ and $h(\boldsymbol{X}^{(*)})$ are direct outputs using the forward pass of the trained model. This procedure is summarized in Algorithm 2 and Algorithm 3. We illustrate the forward pass on the right. This multi-dataset version of our method is later referred to as *CInA (ZS)* (or *Ours (ZS)*).

---

**Algorithm 2** CInA (multi-dataset version).

1: **Input**: Training datasets $\mathbb{D}^{(1)}, ..., \mathbb{D}^{(M)}$.
2: Hyper-parameter: penalty weight $\lambda > 0$.
3: Parameters: $\boldsymbol{\theta}$ (including $\phi$), step size $\eta$.
4: **while** not converged **do**
5:   **for** $m \in [M]$ **do**
6:     Compute $\boldsymbol{K}, \boldsymbol{V}$ using forward pass.
7:     Update $\boldsymbol{\theta} \leftarrow \boldsymbol{\theta} - \eta \nabla \mathcal{L}_{\boldsymbol{\theta}}(\mathbb{D}^{(m)})$.

---

**Algorithm 3** Direct Inference with CInA.

1: **Input**: Test dataset $\mathbb{D}^{(*)}$, trained model, used penalty weight $\lambda$.
2: **Output**: Estimated sample average treatment effect $\hat{\tau}$.
3: Compute $h(\boldsymbol{X}^{(*)}), \boldsymbol{V}^{(*)}$ using forward pass.
4: Compute $\boldsymbol{\alpha}^* = \lambda \cdot \boldsymbol{V}^{(*)}/h(\boldsymbol{X}^{(*)})\boldsymbol{W}^{(*)}$.
5: **return** $\hat{\tau} = (\boldsymbol{\alpha}^* \boldsymbol{W}^{(*)})^\top \boldsymbol{Y}^{(*)}$.

---

**Intuition of What *CInA (ZS)* Learns.** The transformation that encodes for $\boldsymbol{V}$ approximates the solution to the optimization problem in Eq. (3). Thus Algorithm 2 can be seen as learning to how to optimize (Bengio et al., 2021), which enjoys fast inference on a new dataset. It is worth noting that as our optimization problem is continuous and easier to solve than combinatorial optimization, we do not need to employ techniques such as reinforcement learning. We also do not require ground-truth labels to any individual optimization problems as the parameters are learned fully end-to-end.

### 4.3 COMPUTATIONAL COMPLEXITY

We now discuss the computational complexity of our proposed method with respect to the number of units $N$ in each dataset. Suppose the last attention layer uses keys and queries of dimension $D$. Inside each iteration of every epoch, since it needs to compute $\exp(\boldsymbol{k}_i \boldsymbol{k}_j / \sqrt{D})$ for each pair of units $i, j$ and $h(\boldsymbol{X}_i)$ for each $i$, the total complexity of this layer is $\mathcal{O}(N^2 D)$. Based on the outputs of the forward pass, the complexity to evaluate the loss function is $\mathcal{O}(N^2)$, as it evolves computing the penalty term. During inference, the complexity relies on the complexity of the forward pass, as computing $\boldsymbol{\alpha}^*$ and $\hat{\tau}$ are no more than $\mathcal{O}(N)$.

## 5 EXPERIMENTS

We study the performance of CInA on causal inference tasks using both synthetic and real-world datasets. Our objectives are twofold: to validate our theoretical findings in a traditional single-

dataset setting, and to evaluate the feasibility of CInA in a causal foundation modeling context, where the multi-dataset version of CInA will be used for zero-shot causal inference across settings with different levels of difficulty. The detailed implementations of this section can be found in Appendix H. In Appendix I, we provide larger-scale, cross-dataset generalization experiments, as well as two neural baselines (Shi et al., 2019; Chernozhukov et al., 2022).

## 5.1 SIMULATION STUDY A: FIXED CAUSAL GRAPH

**Base Setting.** We follow the simulation study setting in Tarr & Imai (2021), Lee et al. (2010), and Setoguchi et al. (2008) with some modifications. The main purpose of this experiment is to validate our theoretical findings by showing that CInA can perform competitively compared to baselines in the traditional single-dataset setting. We consider a synthetic dataset generated using a fixed causal graph. The covariates of each unit, $X_i$, are drawn from a 10-dimensional multivariate Gaussian distribution with 4 pairs of correlations introduced. Then the treatment is modeled as a single binary variable generated via a logistic model $P(T_i = 1 | X_i) = \text{sigmoid}(\eta^\top h(X_i))$, where $\eta$ is a randomly sampled coefficient parameter, and $h$ is a moderately non-linear and non-additive function detailed in Setoguchi et al. (2008). Finally, the outcome variable is modeled as $Y(T) = \gamma_0 + \gamma^\top x + \tau T + \epsilon$ with $\epsilon \sim \mathcal{N}(0, 0.1)$ and $\tau = -0.4$ (which defines the ATE). For this setting, we generate 100 different datasets sharing the same parameters, each containing 1024 units. We train all baselines, and the *single-dataset version of CInA* in Section 4.1, on each of these 100 datasets *separately*, and evaluate their overall performance. We refer to this setting as the **single-mechanism** setting.

**Variation 1.** In this variation, we aim to evaluate how the multi-dataset version of CInA performs in a zero-shot inference setting with moderate difficulty. We generate 100 different datasets (split into 60/20/20 for training/validation/testing). For each dataset, we first sample a new coefficient parameter $\eta$ from a fixed random distribution $p(\eta)$. We then generate 1024 units using the same form of outcome model specified in the base setting but with a different $\eta$ for each dataset. Our multi-dataset model, *CInA (ZS)*, is trained on 60 training datasets, with hyperparameters selected using 20 validation sets. The evaluation of its zero-shot performance is based on 20 testing datasets. All other baselines are still trained on a dataset-specific manner, i.e., they will be fit to the 20 testing sets separately. We refer to this setting as the **multi-mechanism** setting.

**Variation 2.** In the second variation, similar to variation 1, We generate 100 different datasets, each using a different coefficient parameter $\eta$ from some prior distribution $p(\eta)$. However, instead of sharing the same prior distribution for $\eta$, we force the training/validation datasets and testing datasets to have different supports for $\eta$, i.e., $\text{supp}(p_{\text{training}}(\eta)) = \text{supp}(p_{\text{validation}}(\eta)) \neq \text{supp}(p_{\text{testing}}(\eta))$. We refer to this setting as **multi+OOD**.

**Variation 3.** The third variation is the same as variation 2, except that the 100 datasets have different numbers of units, ranging from $(512, 1024)$. This setting is referred to as **Multi+OOD+diff_size**.

**Baselines (references) and Metrics.** As previous methods are designed for a single dataset, we used them as reference for evaluating our zero-shot method. We consider the following baselines: the *naive* estimator, that performs covariate balancing with uniform weights in $\mathbb{A}$; the *IPW* estimator (Rosenbaum & Rubin, 1983; Rosenbaum, 1987), which performs classical inverse probability weighting with logistic models; the *self-normalized IPW* estimator (Busso et al., 2014; Robins et al., 2007; Imbens, 2004) that normalizes the IPW weights to be in $\mathbb{A}$; the *double machine learning (DML)* estimator (Chernozhukov et al., 2018) with a linear final stage model; and finally, the *SVM* approach which directly solves Eq. (2) as quadratic programming on a per-dataset basis. Among those baselines, the parameter $\lambda$ for SVM was selected using validation datasets,

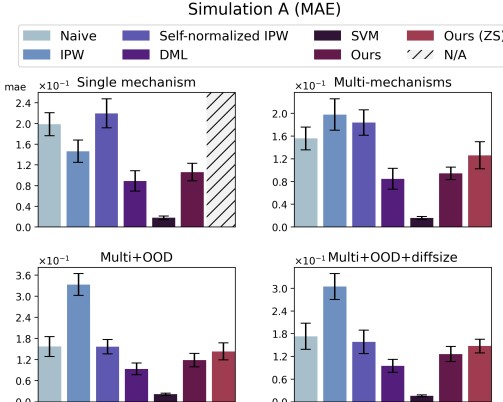

Figure 3: MAE for Simulation A. *CINA* matches the best learning-based method *DML*; *CINA (ZS)* generalizes well in moderate settings.

whenever available. When $\lambda$ is selected properly, the SVM solution should give the exact solution and serve as the ground truth reference for the gradient-based methods, *CInA* and *CInA-(ZS)*. To quantify the accuracy of causal inference, we use mean absolute error (MAE) between true ATE and predicted ATE as the main evaluation metric.

**Results.** Figure 3 shows the results for 4 different settings of simulation A. We observed that across all settings, the single dataset version of *CInA* consistently give on-par performance with *DML*, despite the unfair advantage of *DML* since it utilizes the outcome variables during training. CInA outperforms all other re-weighting based methods except for the ground truth reference, *SVM*. This further confirms the validity of our theoretical findings. Furthermore, in the multi-dataset settings (**Multi-mechanism**, **Multi+OOD** and **Multi+OOD+diff_size**), *CInA (ZS)* shows good zero-shot generalization capabilities under moderate causal mechanism shifts, and performs competitively against other baselines that are trained on the testing datasets themselves on a per-dataset basis.

## 5.2 SIMULATION STUDY B: MULTIPLE CAUSAL GRAPHS

In Section 5.1, we validated our methods in both traditional single-dataset setting and moderate zero-shot settings under the assumption that all tasks/datasets share the same causal graph. Nevertheless, in an ideal context of causal foundational modeling, a good model should be able to perform zero-shot causal inference on datasets coming from both different graphs and different functional relationships. Therefore, in this section, we generate a large number of random synthetic datasets with randomly sampled causal graphs to further evaluate the capability of CInA.

**Datasets.** Following Lachapelle et al. (2019), we generate 5000 datasets (referred to as the **ER-5000** dataset) each using a different random Erdős-Rényi DAG (Erdős & Rényi, 1960). A detailed description is given in Appendix G. All datasets are pre-standardized and split into a 60/20/20 ratio for training/validation/testing. Similar to above, *CInA (ZS)* and *CInA (ZS-S)* (described below) are trained on training datasets, with hyperparameters selected based on validation sets. Reported statistics are based on testing datasets. All baselines are still trained on each testing dataset individually.

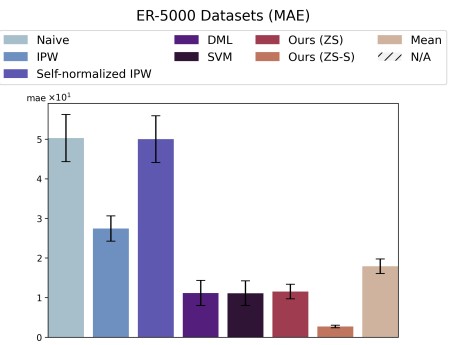

Figure 4: MAEs for ER-5000. *CINA* and *CINA (ZS)* match the best reference method, where *CINA (ZS-S)* improves upon *CINA (ZS)* with additional supervised signals.

**Baselines (references) and Metrics.** The baselines considered in this experiment are the same as Section 5.1, with the exception that the DML baseline performs additional model selection from *linear DML*, *kernel DML*(Nie & Wager, 2021), and *causal forest DML* (Wager & Athey, 2018; Athey et al., 2019). We add another baseline designed for **ER-5000**, dubbed as *mean prediction*, which uses the mean ATE across all training datasets as the prediction for testing datasets. This helps us examine whether CInA is simply memorizing the ATEs from the training set. In addition to the evaluation metric used Section 5.1, we evaluate the computational run-time of all methods on testing datasets.

**Supervised Training of CInA.** Unlike Section 5.1, all datasets in **ER-5000** have different average treatment effects. This allows us to utilize the ground truth ATEs of training datasets as additional supervised signals. We incorporate this via simultaneously minimizing $\sum_{m \in [M]} \left\| (\boldsymbol{V}^{(m)}/h(\boldsymbol{X}^{(m)}))^\top \boldsymbol{Y}^{(m)} - \tau^{(m)} \right\|^2$. The new loss function hence becomes

$$\sum_{m \in [M]} \mathcal{L}_{\boldsymbol{\theta}}(\mathbb{D}^{(m)}) + \mu \sum_{m \in [M]} \left\| (\boldsymbol{V}^{(m)}/h(\boldsymbol{X}^{(m)}))^\top \boldsymbol{Y}^{(m)} - \tau^{(m)} \right\|^2, \tag{8}$$

where $\mu$ is the adjustable coefficient with default value 1. We refer to this supervised variation of our method as *CInA (ZS-S)* (or *Ours (ZS-S)*).

**Results.** Figure 4 summarizes the results on **ER-5000** datasets. We observe that the unsupervised version of *CInA (ZS)* already reached the performance of *DML*, while being able to significantly accelerate the inference computational time by a magnitude of $\sim 10^2$ (Figure 6). With additional supervised signals, *CInA (ZS-S)* is able to significantly outperforms all per-dataset baselines.

## 5.3 EMPIRICAL STUDIES ON REAL-WORLD DATASETS

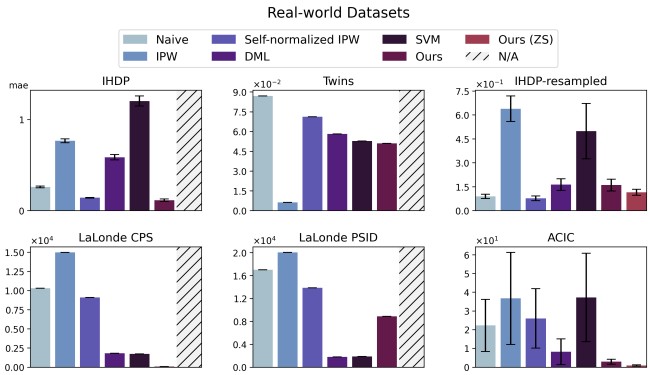

Figure 5: MAE for real-world datasets. *CInA* outperforms the majority of baselines in most cases. *CInA (ZS)* generalizes well and returns the best result for ACIC.

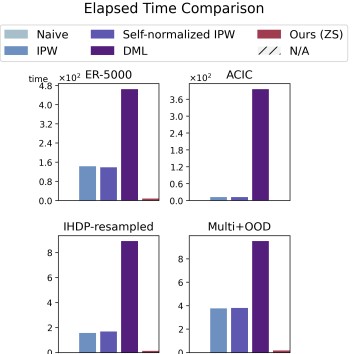

Figure 6: Elapsed time (seconds). *CINA (ZS)* produces estimands instantaneously.

**Datasets and Baselines (references).** We evaluate treatment effect estimation performances on real-world datasets including: **Twins** (Almond et al., 2005), **IHDP** (Hill, 2011), **IHDP-resampled** (Chernozhukov et al., 2022), **ACIC** (Shimoni et al., 2018; MacDorman & Atkinson, 1998), , **LaLonde CPS** and **LaLonde PSID** (LaLonde, 1986). Among them, **IHDP-resampled** and **ACIC** naturally come with multiple datasets, hence can be used to evaluate the zero-shot causal inference for *CInA (ZS)*. For other datasets, only the *single dataset version of CInA* is evaluated due to their single-causal mechanism nature. A detailed description of these datasets can be found in Appendix G. All baselines and cross-validation settings are the same as Section 5.2.

**Results.** Figure 5 summarizes our results. We observe that the experimental findings in simulation studies also hold in real-world settings. In single-dataset experiments, *CInA* is able to outperform the majority of per-dataset baselines in most cases (except for *DML* in **LaLonde PSID** and *IPW* in **Twins**, etc). In multi-dataset experiments, namely, **IHDP-resampled** and **ACIC**, *CInA (ZS)* outperforms the majority of baselines including *CInA*. Furthermore, we noticed that unlike in simulations, *SVM* is not working well in **IHDP-resampled** and **ACIC**. This is potentially because the hyperparameter selection is performed on validation datasets, which by construction, do not represent the causal graphs/functional relationships of the test datasets well. However, our results show that *CInA (ZS)* and *CInA (ZS-S)* are able to robustly perform zero-shot causal inference on unseen datasets in this case. In summary, *CInA* and its variations generally perform well in real-world settings, however its performance may be limited by the availability of dataset resources.

## 6 DISCUSSION

In this work, we take a first step towards building causally-aware foundation models for complex tasks, with a particular focus on the duality between causal inference and attention mechanisms in transformer-based architectures. In theory, we show that covariate balancing can be solved via training any neural network with self-attention as its last layer. Our proposed approach, Causal Inference with Attention (CInA), leverages multiple unlabeled datasets and is capable of performing zero-shot causal inference on unseen data. This stands in contrast to previous approaches, which need to re-optimize on new data. Empirical results show that CInA generalizes well to out-of-distribution datasets and various real-world datasets, reaching and even surpassing the performance of traditional per-dataset causal inference approaches. Therefore, we believe that our methods can serve as a promising building block in the development of causally-aware foundation models.

Going forward, we view it as an important future step to extend the scope of empirical efforts for obtaining a fully pretrained causal foundation model. First, much work remains to be done to build large (public) datasets incorporating large-scale real-world/semi-synthetic data. Second, it would be crucial to improve the efficiency of our method, potentially incorporating techniques from efficient transformers (Child et al., 2019; Kitaev et al., 2020; Katharopoulos et al., 2020; Sun et al., 2023).

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

## A  DISCUSSION ON THE DEFINITION OF (CAUSAL) FOUNDATION MODELS

In this paper, we focus on treatment effect estimation tasks (defined in Section 3.1). Our model is then tailored for generalizable zero-shot estimating average treatment effects. That is, given unseen datasets/contexts that contains observational records of covariates, treatments, and effects, we aim to estimate the underlying treatment effects using a forward pass of the underlying model.

This approach is inline with the definition of foundation models discussed in Bommasani et al. (2021): "any model that is trained on broad data (generally using self-supervision at scale) that can be adapted (e.g., fine-tuned) to a wide range of downstream tasks". Note that such *task-universality* of foundation models does not necessarily imply adaptability across different *machine learning formulations* (e.g., prediction, imputation, ATE, CATE, counterfactuals); instead, it can refer to adaptability across different *contexts* for a given task. This perspective is widely embraced by recent studies, such as those focusing on foundation models for tabular datasets (Zhang et al., 2023b), time series (Garza & Mergenthaler-Canseco, 2023; Das et al., 2023), and knowledge graphs (Galkin et al., 2023). These studies concentrate exclusively on a single type of task, but assess in-context generalization across datasets.

## B  EXTENDED RELATED WORKS

As our work also intersects with the literature on neural causal estimation methods, we provide a discussion in this section.

**Neural Estimation Methods for Treatment Effects.** Research in this direction employs deep learning methods to estimate treatment effects, typically relying on standard assumptions that ensure identifiability, similar to our setting. A prominent approach focuses on learning a representation of the covariates that is predictive of the outcome (Johansson et al., 2016; Shalit et al., 2017; Yao et al., 2018). Following this, several methods have been proposed to combine outcome models learned through neural networks with balanced propensity weights (Alaa et al., 2017; Schwab et al., 2018; Du et al., 2021). Semi-parameteric estimation theory and doubly robust estimators have also been applied in neural estimation methods, e.g., using regularization (Shi et al., 2019) or shared representations (Chernozhukov et al., 2018). Another perspective of using neural network is to control for complex relationships and covariates. Kallus (2020a) extends adversarial covariate balancing (Kallus, 2020b) using flexible modeling with neural networks. Generative causal models have also been proposed to leverage the expressivity of neural networks to approximate structural causal models (Louizos et al., 2017; Kocaoglu et al., 2017; Alaa & Van Der Schaar, 2017; Yoon et al., 2018; Pawlowski et al., 2020; Xia et al., 2021; 2022), which then allows for the estimation of treatment effects. In addition, Xia et al. (2021) also proved that their proposed method can be used to test the identifiability of causal effect in terms of do-interventions (Pearl, 2009) in the general setting. Xia et al. (2022) extended such testing for counterfactual outcomes (Bareinboim et al., 2022). In (Melnychuk et al., 2022), the attention mechanism was employed to estimate treatment effect over time for a given unit. Concurrent to our work, Nilforoshan et al. (2023) proposed a meta-learning framework to learn causal effects of various structured treatments on the same population. Their method leverages information across different treatments, which allows for zero-shot learning on an unseen treatment. Our work can be viewed as orthogonal, as we focus on learning the causal effects of the same treatment across different populations.

## C  OMITTED PROOFS

### C.1  DERIVATIONS OF EQ. (1) AND EQ. (2)

We first establish the conditional bias decomposition:

$$\mathbb{E}\left(\hat{\tau} - \tau_{SATE} \mid \{\boldsymbol{X}_i, T_i\}_{i=1}^N\right)$$
$$= \mathbb{E}\left(\sum_{i=1}^N \alpha_i W_i Y_i - \sum_{i=1}^N \frac{1}{N}\big(Y_i(1) - Y_i(0)\big) \mid \{\boldsymbol{X}_i, T_i\}_{i=1}^N\right)$$

$$= \sum_{i=1}^{N} \alpha_i W_i \mathbb{E}\left(Y_i(T_i) \mid \boldsymbol{X}_i, T_i\right) + \sum_{i=1}^{N} \frac{1}{N} \mathbb{E}\left(Y_i(1) - Y_i(0) \mid \boldsymbol{X}_i, T_i\right)$$

$$= \sum_{i=1}^{N} \left(\alpha_i W_i \mathbb{E}\left(Y_i(0) \mid \boldsymbol{X}_i\right) + \alpha_i T_i \mathbb{E}\left(Y_i(1) - Y_i(0) \mid \boldsymbol{X}_i\right)\right) + \sum_{i=1}^{N} \frac{1}{N} \mathbb{E}\left(Y_i(1) - Y_i(0) \mid \boldsymbol{X}_i\right)$$

$$= \sum_{i=1}^{N} (\alpha_i T_i - \frac{1}{N}) \mathbb{E}\left(Y_i(1) - Y_i(0) \mid \boldsymbol{X}_i\right) + \sum_{i=1}^{N} \alpha_i W_i \mathbb{E}\left(Y_i(0) \mid \boldsymbol{X}_i\right),$$

where we use the assumption of consistency between observed and potential outcomes and non-interference between unit (SUTVA, Rubin (1990)) in the second equation and unconfoundedness in the third equation.

Formally, define a feature map $\phi : \mathbb{X} \to \mathcal{H}_\phi$, where $\mathbb{X}$ is the support of covariates and $\mathcal{H}_\phi$ is some Hilbert space. The unit-ball RKHS is given by $\mathcal{F}_\phi = \{f : \mathbb{X} \to \mathbb{R} \mid \exists \theta \in \mathcal{H}_\phi,\ s.t.\ f(x) = \langle \theta, \phi(x) \rangle,\ \forall x \in \mathbb{X}\ and\ \|\theta\| \le 1\}$. Recall that $\langle \cdot, \cdot \rangle$ denotes the inner product of Hilbert space $\mathcal{H}_\phi$ and $\|\cdot\|$ denotes the associated norm. The adversarial upper bound of the square of the second term in the conditional bias can be calculated via

$$\sup_{f \in \mathcal{F}_\phi} \left(\sum_{i=1}^{N} \alpha_i W_i f(\boldsymbol{X}_i)\right)^2$$

$$= \sup_{\theta \in \mathcal{H}_\phi, \|\theta\| \le 1} \left(\sum_{i=1}^{N} \alpha_i W_i \left\langle \theta, \phi(\boldsymbol{X}_i)\right\rangle\right)^2$$

$$= \sup_{\theta \in \mathcal{H}_\phi, \|\theta\| \le 1} \left(\left\langle \theta, \sum_{i=1}^{N} \alpha_i W_i \phi(\boldsymbol{X}_i)\right\rangle\right)^2$$

$$\le \left\|\sum_{i=1}^{N} \alpha_i W_i \phi(\boldsymbol{X}_i)\right\|^2 = \boldsymbol{\alpha}^\top \boldsymbol{K}_\phi \boldsymbol{\alpha}.$$

Recall that $[\boldsymbol{K}_\phi]_{ij} = W_i W_j \langle \phi(\boldsymbol{X}_i), \phi(\boldsymbol{X}_j) \rangle$. Therefore minimizing this adversarial loss subject to $\boldsymbol{\alpha} \in \mathbb{A}$ reduces to Eq. (1).

By evoking Theorem 1 in Tarr & Imai (2021), we have that Eq. (1) is equivalent to Eq. (2) for some $\lambda \ge 0$. However, the exact value of $\lambda$ depends on $\boldsymbol{K}_\phi$. For example, if $\boldsymbol{K}_\phi$ is such that the minimum value of Eq. (1) is 0, then $\lambda = 0$. This is because the minimizer of Eq. (1) would also be the minimizer under the unnormalized constraint (Eq. (2) with $\lambda = 0$), as $\boldsymbol{\alpha}^\top \boldsymbol{K}_\phi \boldsymbol{\alpha} \ge 0$ for any $\boldsymbol{\alpha} \in \mathbb{R}^N$.

Conversely, we can also show that $\lambda > 0$ if $\boldsymbol{K}_\phi$ is of full rank.

**Lemma 1.** *If $\boldsymbol{K}_\phi$ if of full rank, then $\lambda > 0$.*

*Proof.* From the proof of Theorem 1 in Tarr & Imai (2021), we know that $\lambda = 0$ only if $q_* = \min_{\boldsymbol{W}^\top \boldsymbol{\alpha} = 0, \boldsymbol{0} \preceq \boldsymbol{\alpha} \preceq \boldsymbol{1}, \boldsymbol{\alpha} \ne 0} \frac{\sqrt{\boldsymbol{\alpha}^\top \boldsymbol{K}_\phi \boldsymbol{\alpha}}}{\boldsymbol{1}^\top \boldsymbol{\alpha}/2}$ is zero. However, since $\boldsymbol{K}_\phi$ is of full rank, it is positive definite. Thus for any $\boldsymbol{\alpha} \ne 0$, there is $\boldsymbol{\alpha}^\top \boldsymbol{K}_\phi \boldsymbol{\alpha} > 0$. Therefore $q_* > 0$. Consequently, $\lambda > 0$. $\square$

### C.2 DERIVATIONS OF EQ. (3) AND EQ. (4)

The dual form of Eq. (3) can be derived using its Lagrangian

$$L(\boldsymbol{\beta}, \beta_0, \boldsymbol{\xi}, \boldsymbol{\alpha}, \bar{\boldsymbol{\alpha}}) = \frac{\lambda}{2}\|\boldsymbol{\beta}\|^2 + \sum_{i=1}^{N} \xi_i + \sum_{i=1}^{N} \alpha_i \left(1 - \xi_i - W_i\left(\langle \boldsymbol{\beta}, \phi(\boldsymbol{X}_i) \rangle + \beta_0\right)\right) - \sum_{i=1}^{N} \bar{\alpha}_i \xi_i,$$

where $\boldsymbol{\alpha} \succeq \boldsymbol{0}$ and $\bar{\boldsymbol{\alpha}} \succeq \boldsymbol{0}$. The primal form in Eq. (3) can be obtained by $\min_{\boldsymbol{\beta}, \beta_0, \xi_i} \max_{\boldsymbol{\alpha} \succeq \boldsymbol{0}, \bar{\boldsymbol{\alpha}} \succeq \boldsymbol{0}} L(\boldsymbol{\beta}, \beta_0, \boldsymbol{\xi}, \boldsymbol{\alpha}, \bar{\boldsymbol{\alpha}})$. If we exchange $\min \max$ with $\max \min$, solving

$\min_{\boldsymbol{\beta},\beta_0,\xi_i}$ by setting the derivatives to zero leads to

$$\nabla_{\boldsymbol{\beta}} L(\boldsymbol{\beta},\beta_0,\xi,\boldsymbol{\alpha},\bar{\boldsymbol{\alpha}}) = \lambda\boldsymbol{\beta} - \sum_{i=1}^{N} \alpha_i W_i \phi(\boldsymbol{X}_i) = \mathbf{0},$$

$$\nabla_{\beta_0} L(\boldsymbol{\beta},\beta_0,\xi,\boldsymbol{\alpha},\bar{\boldsymbol{\alpha}}) = -\sum_{i=1}^{N} \alpha_i W_i = 0,$$

$$\nabla_{\xi_i} L(\boldsymbol{\beta},\beta_0,\xi,\boldsymbol{\alpha},\bar{\boldsymbol{\alpha}}) = 1 - \alpha_i - \bar{\alpha}_i = 0, \ \forall\, i \in [N].$$

Plugging these in $L(\boldsymbol{\beta},\beta_0,\xi,\boldsymbol{\alpha},\bar{\boldsymbol{\alpha}})$, we can reduce $\max_{\boldsymbol{\alpha}\succeq\mathbf{0},\bar{\boldsymbol{\alpha}}\succeq\mathbf{0}}\min_{\boldsymbol{\beta},\beta_0,\xi_i} L(\boldsymbol{\beta},\beta_0,\boldsymbol{\xi},\boldsymbol{\alpha},\bar{\boldsymbol{\alpha}})$ to Eq. (2). Thus it is the dual form of Eq. (3).

In addition, we can also derive Eq. (4). It is easy to check that Slater's condition holds for the primal SVM problem in Eq. (3). Thus it satisfies strong duality. Therefore any optimal solutions to the primal-dual problems must satisfy the KKT condition $\lambda\boldsymbol{\beta}^* = \sum_{j=1}^{N} \alpha_j^* W_j \phi(\boldsymbol{X}_j)$.

### C.3 DERIVATIONS OF EQ. (6)

From the Taylor expansion

$$\exp(\boldsymbol{k}_i^\top \boldsymbol{k}_j/\sqrt{D}) = \sum_{l=0}^{+\infty} \frac{1}{l!}(\boldsymbol{k}_i^\top \boldsymbol{k}_j/\sqrt{D})^l$$

$$= \sum_{l=0}^{+\infty} \sum_{N_1+\ldots+N_D=l} \frac{\big([\boldsymbol{k}_i]_1^{N_1}\ldots[\boldsymbol{k}_i]_D^{N_D}\big)\big([\boldsymbol{k}_j]_1^{N_1}\ldots[\boldsymbol{k}_j]_D^{N_D}\big)}{D^{l/2} N_1!\ldots N_D!},$$

we have that $\exp(\boldsymbol{k}_i^\top \boldsymbol{k}_j/\sqrt{D}) = \langle\phi(\boldsymbol{X}_i),\phi(\boldsymbol{X}_j)\rangle$ if

$$\phi(\boldsymbol{x}) = \left(\frac{[\boldsymbol{k}]_1^{N_1}\ldots[\boldsymbol{k}]_D^{N_D}}{D^{l/2}(N_1!\ldots N_D!)^{1/2}}\right)_{N_1+\ldots+N_D=l,\ l\in\mathbb{N}}. \tag{9}$$

Here $\boldsymbol{k}$ denotes the key embedding of $\boldsymbol{x}$ following the same transformation that $\boldsymbol{k}_i$ is obtained from $\boldsymbol{X}_i$. Note that we allow the transformation to depend on $\boldsymbol{X}$, which corresponds to a data-dependent kernel.

Using this expression, the $i$-th output of the self-attention layer when $\boldsymbol{Q} = \boldsymbol{K}$ can be equivalently written as

$$\sum_{j=1}^{N} \frac{\exp\big(\boldsymbol{k}_i^\top \boldsymbol{k}_j/\sqrt{D}\big)}{\sum_{j'=1}^{N}\exp\big(\boldsymbol{k}_i^\top \boldsymbol{k}_{j'}/\sqrt{D}\big)} v_j = \sum_{j=1}^{N} \frac{\langle\phi(\boldsymbol{X}_i),\phi(\boldsymbol{X}_j)\rangle}{h(\boldsymbol{X}_i)} v_i = \sum_{j=1}^{N} \frac{v_j}{h(\boldsymbol{X}_j)}\langle\phi(\boldsymbol{X}_j),\phi(\boldsymbol{X}_i)\rangle.$$

### C.4 PROOF OF THEOREM 1

We first state its formal version:

**Theorem 1.** *If the covariates $\boldsymbol{X}$ satisfy that $\phi(\boldsymbol{X}_1),...,\phi(\boldsymbol{X}_N)$ are linearly independent, then Algorithm 1 recovers the optimal balancing weight at the global minimum of the penalized hinge loss in Eq. (7).*

*In particular, the optimal solution $\boldsymbol{\alpha}^*$ to Eq. (1), in which the feature function $\phi$ is defined using the optimal neural network parameters via Eq. (9), can be obtained using the optimal neural network parameters that minimize Eq. (7) via $\alpha_j^* = \lambda v_j/h(\boldsymbol{X}_j)W_j$.*

*Proof.* Denote $\boldsymbol{\beta} = \sum_{j=1}^{N} \frac{v_j}{h(\boldsymbol{X}_j)}\phi(\boldsymbol{X}_j)$, then using Eq. (6), we can rewrite the loss function in Eq. (7) as

$$\mathcal{L}_{\boldsymbol{\theta}}(\mathbb{D}) = \frac{\lambda}{2}\|\boldsymbol{\beta}\|^2 + \sum_{i=1}^{N} \big[1 - W_i\big(\langle\boldsymbol{\beta},\phi(\boldsymbol{X}_i)\rangle + \beta_0\big)\big]_+.$$

Denote $\xi_i = \left[1 - W_i\big(\langle \boldsymbol{\beta}, \phi(\boldsymbol{X}_i)\rangle + \beta_0\big)\right]_+$, then minimizing $\mathcal{L}_{\boldsymbol{\theta}}(\mathbb{D})$ can be equivalently written as

$$
\begin{aligned}
\min_{\boldsymbol{\theta}} \quad & \frac{\lambda}{2}\|\boldsymbol{\beta}\|^2 + \sum_{i=1}^{N} \xi_i, \\
s.t. \quad & W_i\left(\langle \boldsymbol{\beta}, \phi(\boldsymbol{X}_i)\rangle + \beta_0\right) \geq 1 - \xi_i, \quad \xi_i \geq 0, \quad \forall i \in [N].
\end{aligned}
$$

Thus at the optimal $\boldsymbol{\theta}$, the corresponding $\boldsymbol{\beta}$ is also the optimal solution to

$$
\begin{aligned}
\min_{\boldsymbol{\beta}, \beta_0, \boldsymbol{\xi}} \quad & \frac{\lambda}{2}\|\boldsymbol{\beta}\|^2 + \sum_{i=1}^{N} \xi_i, \\
s.t. \quad & W_i\left(\langle \boldsymbol{\beta}, \phi(\boldsymbol{X}_i)\rangle + \beta_0\right) \geq 1 - \xi_i, \quad \xi_i \geq 0, \quad \forall i \in [N],
\end{aligned}
$$

where $\phi$ is defined using the optimal $\boldsymbol{\theta}$. This recovers the primal SVM problem. By the primal-dual connection proven in Appendix C.2, if we denote the optimal solution to the dual problem (which is Eq. (2)) as $\boldsymbol{\alpha}^*$, we have

$$
\lambda \boldsymbol{\beta} = \sum_{j=1}^{N} \alpha_j^* W_j \phi(\boldsymbol{X}_j).
$$

Consequently, by the definition of $\boldsymbol{\beta}$, we have

$$
\sum_{j=1}^{N} \frac{\lambda v_j}{h(\boldsymbol{X}_j)} \phi(\boldsymbol{X}_j) = \sum_{j=1}^{N} \alpha_j^* W_j \phi(\boldsymbol{X}_j).
$$

By the assumption that $\phi(\boldsymbol{X}_1), ..., \phi(\boldsymbol{X}_N)$ are linearly independent, we must have $\frac{\lambda v_j}{h(\boldsymbol{X}_j)} = \alpha_j^* W_j$ for all $j \in [N]$. Therefore $\alpha_j^* = \lambda v_j / h(\boldsymbol{X}_j) W_j$. $\qquad \square$

**Remark 1.** *Note that when $\phi(\boldsymbol{X}_1), ..., \phi(\boldsymbol{X}_N)$ are linearly independent, the matrix $\boldsymbol{K}_\phi = [W_1\phi(\boldsymbol{X}_1), ..., W_N\phi(\boldsymbol{X}_N)]^\top [W_1\phi(\boldsymbol{X}_1), ..., W_N\phi(\boldsymbol{X}_N)]$ is of full rank. Thus by Lemma 1, there is $\lambda > 0$. Conversely, using a similar decomposition, we know that if $\hat{\boldsymbol{K}}_\phi = [\phi(\boldsymbol{X}_1), ..., \phi(\boldsymbol{X}_N)]^\top [\phi(\boldsymbol{X}_1), ..., \phi(\boldsymbol{X}_N)]$ is of full rank, then $\phi(\boldsymbol{X}_1), ..., \phi(\boldsymbol{X}_N)$ are linearly independent. Since $\hat{\boldsymbol{K}}_\phi = \exp(\boldsymbol{K}\boldsymbol{K}^\top / \sqrt{D})$, we have $\phi(\boldsymbol{X}_1), ..., \phi(\boldsymbol{X}_N)$ linearly independent if $\boldsymbol{K}$ is of row rank $N$. Thus the assumption on $\boldsymbol{X}$ in Theorem 1 is satisfied when $\boldsymbol{K}$ is of row rank $N$.*

## D    ALTERNATIVE OBJECTIVES

Consider minimizing the square of both terms in the conditional bias, which we decompose into the following form

$$
\begin{aligned}
& \left(\mathbb{E}\left(\hat{\tau} - \tau_{SATE} \mid \{\boldsymbol{X}_i, T_i\}_{i=1}^N\right)\right)^2 \\
&= \left(\sum_{i=1}^{N} \alpha_i W_i \mathbb{E}\left(Y_i(T_i)|\boldsymbol{X}_i, T_i\right) - \frac{1}{N}\sum_{i=1}^{N}\left(\mathbb{E}\left(Y_i(1)|\boldsymbol{X}_i\right) - \mathbb{E}\left(Y_i(0)|\boldsymbol{X}_i\right)\right)\right)^2.
\end{aligned} \tag{10}
$$

Denote the outcome models $\mathbb{E}(Y_i(1)|\boldsymbol{X}_i) = f_1(\boldsymbol{X}_i)$ and $\mathbb{E}(Y_i(0)|\boldsymbol{X}_i) = f_0(\boldsymbol{X}_i)$. We choose to minimize the above term in worst case over all possible potential outcome models $(f_0, f_1) \in \mathcal{F}_\phi^2$. Here the space $\mathcal{F}_\phi^2$ is defined as $\mathcal{F}_\phi^2 = \{(f_0, f_1) \mid f_0 \in \mathcal{F}_\phi, f_1 \in \mathcal{F}_\phi\}$.

Suppose $f_0(x) = \langle \phi(x), \theta_0 \rangle$ and $f_1(x) = \langle \phi(x), \theta_1 \rangle$ for $\theta_0, \theta_1 \in \mathcal{H}_\phi, \|\theta_0\| \leq 1, \|\theta_1\| \leq 1$. We can bound Eq. (10) with respect to all outcome models in $\mathcal{F}_\phi^2$ as

$$
\left( \sum_{i=1}^N \alpha_i W_i f_{T_i}(\boldsymbol{X}_i) - \frac{1}{N} \sum_{i=1}^N \left( f_1(\boldsymbol{X}_i) - f_0(\boldsymbol{X}_i) \right) \right)^2
$$

$$
= \left( \left\langle \sum_{i \in \mathbb{T}} \alpha_i W_i \phi(\boldsymbol{X}_i) - \frac{1}{N} \sum_{i \in [N]} \phi(\boldsymbol{X}_i), \theta_1 \right\rangle + \left\langle \sum_{i \in \mathbb{C}} \alpha_i W_i \phi(\boldsymbol{X}_i) + \frac{1}{N} \sum_{i \in [N]} \phi(X_i), \theta_0 \right\rangle \right)^2
$$

$$
\leq 2 \left( \sum_{i \in \mathbb{T}} \alpha_i W_i \phi(\boldsymbol{X}_i) - \frac{1}{N} \sum_{i \in [N]} \phi(\boldsymbol{X}_i) \right)^2 + 2 \left( \sum_{i \in \mathbb{C}} \alpha_i W_i \phi(\boldsymbol{X}_i) + \frac{1}{N} \sum_{i \in [N]} \phi(\boldsymbol{X}_i) \right)^2
$$

where the inequality uses Cauchy-Schwartz inequality. Minimizing this upper bound subject to $\boldsymbol{\alpha} \in \mathbb{A}$ is equivalent to solving

$$
\begin{aligned}
\min_{\boldsymbol{\alpha}} \quad & \boldsymbol{\alpha}^\top \boldsymbol{G}_\phi \boldsymbol{\alpha} + \boldsymbol{\alpha}^\top \boldsymbol{g}_\phi, \\
s.t. \quad & \sum_{i \in \mathbb{T}} \alpha_i = \sum_{i \in \mathbb{C}} \alpha_i = 1, \quad \boldsymbol{0} \preceq \boldsymbol{\alpha} \preceq \boldsymbol{1}.
\end{aligned}
\tag{11}
$$

Here

$$
[\boldsymbol{G}_\phi]_{i,j} = \delta_{W_i = W_j} \langle \phi(\boldsymbol{X}_i), \phi(\boldsymbol{X}_j) \rangle,
$$

$$
[\boldsymbol{g}_\phi]_i = -\frac{2}{N} \sum_{j=1}^N \langle \phi(\boldsymbol{X}_i), \phi(\boldsymbol{X}_i) \rangle.
$$

It is easy to show that $\boldsymbol{G}_\phi \succeq 0$ as it can be decomposed into two submatrixes which are positive semi-definite. In addition, as $\langle \phi(\boldsymbol{X}_i), \phi(\boldsymbol{X}_j) \rangle = \exp(\boldsymbol{k}_i^\top \boldsymbol{k}_j / \sqrt{D}) > 0$, we know that $\boldsymbol{g}_\phi \prec \boldsymbol{0}$.

To come up with a consistent gradient-based solver, notice first that Eq. (11) is equivalent to the following unnormalized problem for some $\lambda, \mu \geq 0$

$$
\begin{aligned}
\min_{\boldsymbol{\alpha}} \quad & \boldsymbol{\alpha}^\top \boldsymbol{G}_\phi \boldsymbol{\alpha} + 2\mu \cdot \boldsymbol{g}_\phi^\top \boldsymbol{\alpha} - 2\lambda \cdot \boldsymbol{1}^\top \boldsymbol{\alpha}, \\
s.t. \quad & \boldsymbol{W}^\top \boldsymbol{\alpha} = 0, \quad \boldsymbol{0} \preceq \boldsymbol{\alpha} \preceq \boldsymbol{1}.
\end{aligned}
\tag{12}
$$

This can be shown similarly to the proof of Theorem 1 in Tarr & Imai (2021). We escape the details but provide the following main steps:

1. We first show that for some $\epsilon_\lambda, \epsilon_\mu \geq 0$, Eq. (12) is equivalent to

$$
\begin{aligned}
\min_{\boldsymbol{\alpha}} \quad & \boldsymbol{\alpha}^\top \boldsymbol{G}_\phi \boldsymbol{\alpha}, \\
s.t. \quad & \boldsymbol{W}^\top \boldsymbol{\alpha} = 0, \quad \boldsymbol{0} \preceq \boldsymbol{\alpha} \preceq \boldsymbol{1}, \quad -\boldsymbol{g}_\phi^\top \boldsymbol{\alpha} \geq \epsilon_\mu, \quad \boldsymbol{1}^\top \boldsymbol{\alpha} \geq \epsilon_\lambda.
\end{aligned}
$$

2. Next, we show that the above problem is equivalent to

$$
\begin{aligned}
\min_{\boldsymbol{\alpha}} \quad & \sqrt{\boldsymbol{\alpha}^\top \boldsymbol{G}_\phi \boldsymbol{\alpha}}, \\
s.t. \quad & \boldsymbol{W}^\top \boldsymbol{\alpha} = 0, \quad \boldsymbol{0} \preceq \boldsymbol{\alpha} \preceq \boldsymbol{1}, \quad -\boldsymbol{g}_\phi^\top \boldsymbol{\alpha} \geq \epsilon_\mu, \quad \boldsymbol{1}^\top \boldsymbol{\alpha} \geq \epsilon_\lambda,
\end{aligned}
$$

which is equivalent to

$$
\begin{aligned}
\min_{\boldsymbol{\alpha}} \quad & \sqrt{\boldsymbol{\alpha}^\top \boldsymbol{G}_\phi \boldsymbol{\alpha}} + \nu_\mu \cdot \boldsymbol{g}_\phi^\top \boldsymbol{\alpha} - \nu_\lambda \boldsymbol{1}^\top \boldsymbol{\alpha}, \\
s.t. \quad & \boldsymbol{W}^\top \boldsymbol{\alpha} = 0, \quad \boldsymbol{0} \preceq \boldsymbol{\alpha} \preceq \boldsymbol{1}.
\end{aligned}
$$

for some $\nu_\lambda, \nu_\mu \geq 0$.

3. For some $\lambda \geq 0$, the above problem is equivalent to

$$\min_{\boldsymbol{\alpha}} \quad \frac{\sqrt{\boldsymbol{\alpha}^\top \boldsymbol{G}_\phi \boldsymbol{\alpha}} + \nu_\mu \cdot \boldsymbol{g}_\phi^\top \boldsymbol{\alpha}}{\mathbf{1}^\top \boldsymbol{\alpha}},$$
$$s.t. \quad \boldsymbol{W}^\top \boldsymbol{\alpha} = 0, \quad \mathbf{0} \preceq \boldsymbol{\alpha} \preceq \mathbf{1}.$$

Since this problem is scale-free, it is equivalent to

$$\min_{\boldsymbol{\alpha}} \quad \frac{\sqrt{\boldsymbol{\alpha}^\top \boldsymbol{G}_\phi \boldsymbol{\alpha}} + \nu_\mu \cdot \boldsymbol{g}_\phi^\top \boldsymbol{\alpha}}{\mathbf{1}^\top \boldsymbol{\alpha}},$$
$$s.t. \quad \sum_{i \in \mathbb{T}} \alpha_i = \sum_{i \in \mathbb{C}} \alpha_i = 1, \quad \mathbf{0} \preceq \boldsymbol{\alpha} \preceq \mathbf{1},$$

i.e.,

$$\min_{\boldsymbol{\alpha}} \quad \sqrt{\boldsymbol{\alpha}^\top \boldsymbol{G}_\phi \boldsymbol{\alpha}} + \nu_\mu \cdot \boldsymbol{g}_\phi^\top \boldsymbol{\alpha},$$
$$s.t. \quad \sum_{i \in \mathbb{T}} \alpha_i = \sum_{i \in \mathbb{C}} \alpha_i = 1, \quad \mathbf{0} \preceq \boldsymbol{\alpha} \preceq \mathbf{1},$$

4. Using similar arguments as above, one can show the above problem is equivalent to

$$\min_{\boldsymbol{\alpha}} \quad \boldsymbol{\alpha}^\top \boldsymbol{G}_\phi \boldsymbol{\alpha} + \boldsymbol{g}_\phi^\top \boldsymbol{\alpha},$$
$$s.t. \quad \sum_{i \in \mathbb{T}} \alpha_i = \sum_{i \in \mathbb{C}} \alpha_i = 1, \quad \mathbf{0} \preceq \boldsymbol{\alpha} \preceq \mathbf{1},$$

for some $\mu \geq 0$.

The primal form of Eq. (12) can be written as

$$\min_{\boldsymbol{\beta}_1, \boldsymbol{\beta}_2, \beta_0, \boldsymbol{\xi}} \quad \frac{1}{2}\|\boldsymbol{\beta}_1\|^2 + \frac{1}{2}\|\boldsymbol{\beta}_2\|^2 + \sum_{i=1}^N \xi_i,$$
$$s.t. \quad \left(\langle \boldsymbol{\beta}_1, \phi(\boldsymbol{X}_i)\rangle + \beta_0\right) \geq \lambda - \mu[\boldsymbol{g}_\phi]_i - \xi_i, \quad \forall i \in \mathbb{T}$$
$$\left(\langle \boldsymbol{\beta}_2, \phi(\boldsymbol{X}_i)\rangle - \beta_0\right) \geq \lambda - \mu[\boldsymbol{g}_\phi]_i - \xi_i, \quad \forall i \in \mathbb{C}$$
$$\xi_i \geq 0, \quad \forall i \in [N].$$

Following similar derivations in Appendix C, we can write out an unconstrained loss function

$$\mathcal{L}_{\boldsymbol{\theta}}(\mathbb{D}) = \frac{1}{2}\left\|\sum_{j \in \mathbb{T}} \frac{v_j}{h(\boldsymbol{X}_j)}\phi(\boldsymbol{X}_j)\right\|^2 + \frac{1}{2}\left\|\sum_{j \in \mathbb{C}} \frac{v_j}{h(\boldsymbol{X}_j)}\phi(\boldsymbol{X}_j)\right\|^2$$
$$+ \left[\lambda - \mu[\boldsymbol{g}_\phi]_\mathbb{T} - \left(\mathrm{softmax}(\boldsymbol{K}_\mathbb{T}\boldsymbol{K}_\mathbb{T}^\top/\sqrt{D})\boldsymbol{V}_\mathbb{T} + \beta_0\right)\right]_+$$
$$+ \left[\lambda - \mu[\boldsymbol{g}_\phi]_\mathbb{C} - \left(\mathrm{softmax}(\boldsymbol{K}_\mathbb{C}\boldsymbol{K}_\mathbb{C}^\top/\sqrt{D})\boldsymbol{V}_\mathbb{C} - \beta_0\right)\right]_+,$$

where the optimal $\boldsymbol{\alpha}^*$ solving Eq. (11) can be read off as $\alpha_i = \frac{v_i}{h(\boldsymbol{X}_i)}$.

For the conditional mean square error, under regularity constraints in Bennett & Kallus (2019), we can also use the same upper bound as above (up to an additive $\mathcal{O}(1/N)$ gap). Therefore the same derivation holds. However, as this loss function separates the treated group from the control group aside from sharing the constant intercept $\beta_0$, it might not be preferable than the objective proposed in the main text.

## E  NON-BINARY TREATMENTS

Consider a generalization to the setting in Section 3.1, where the dataset $\mathbb{D} = \{(\boldsymbol{X}_i, \boldsymbol{T}_i, Y_i)\}_{i \in [N]}$ in which $\boldsymbol{T}_i$ is a $S$-dimensional vector of multiple binary treatments. Let $Y_i^s(t)$ be the potential outcome of assigning treatment $[\boldsymbol{T}_i]_s = t$.

Assuming SUTVA ($Y_i = Y_i^s([\boldsymbol{T}_i]_s)$) and unconfoundedness. Denote $\mathbb{T}^s = \{i \in [N] : [\boldsymbol{T}_i]_s = 1\}$ and $\mathbb{C}^s = \{i \in [N] : [\boldsymbol{T}_i]_s = 0\}$. We consider weighted estimators in the form of

$$\hat{\tau}^s = \sum_{i \in \mathbb{T}^s} \alpha_i Y_i^s(1) - \sum_{i \in \mathbb{C}^s} \alpha_i Y_i^s(0)$$

for the sample average treatment of the $s$-th treatment

$$\tau_{SATE}^s = \frac{1}{N} \sum_{i=1}^N \left( Y_i^s(1) - Y_i^s(0) \right).$$

Following the same derivations in Section 3 and Appendix C, we can obtain a dual-SVM formulation to optimize $\boldsymbol{\alpha}$ in the adversarial case. This dual-SVM formulation can then be transformed into its primal problem. As self-attention is implicitly implementing the predictor in the primal problem, we can then read off the optimal $\boldsymbol{\alpha}^*$ by training this self-attention-based neural network with a penalized hinge loss.

However, as we would like to evaluate the sample average treatment for multiple treatments, we can actually aggregate $S$ SVM problems together using the flexibility of self-attention layers. Namely, instead of consider a one-dimensional value vector $\boldsymbol{V}$ in Section 3.2, we use $\boldsymbol{V} \in \mathbb{R}^{N \times S}$, where the $s$-th dimension corresponds to the $s$-th treatment. By minimizing the following loss function

$$\mathcal{L}_{\boldsymbol{\theta}}(\mathbb{D}) = \frac{\lambda}{2} \sum_{s=1}^S \left\| \sum_{j=1}^N \frac{[\boldsymbol{V}]_{js}}{h(\boldsymbol{X}_j)} \phi(\boldsymbol{X}_j) \right\|^2 + \sum_{s=1}^S \left[ \boldsymbol{1} - \boldsymbol{W}_{:,s} \left( \text{softmax}(\boldsymbol{K}\boldsymbol{K}^\top / \sqrt{D}) \boldsymbol{V}_{:,s} + \beta_0 \right) \right]_+,$$

we can read off the optimal balancing weight $\boldsymbol{\alpha}$ for the $s$-th treatment via $\lambda \cdot \boldsymbol{V}_{:,s} / h(\boldsymbol{X}) \boldsymbol{W}_{:,s}$

# F  INDIVIDUAL TREATMENT EFFECT ESTIMATION

In this section, we further consider the problem of estimating individual treatment effect (ITE) in the binary treatment setup of Section 3. Here we present one possible algorithmic approach to approximate ITEs with CInA. Without loss of generality, suppose $T_1 = 1$ and we would like to estimate ITE on the first unit $\mathbb{E}(Y_1(1) - Y_1(0) \mid \boldsymbol{X}_1)$.

Denote the "counterfactual dataset" by replacing the first sample with $(\boldsymbol{X}_1, 0, \hat{Y}_1(0))$ as $\hat{\mathbb{D}}$, where $\hat{Y}_1(0)$ is a realization of $Y_1(0)$. Note that we do not have access to the value of $\hat{Y}_1(0)$. However, we do have access to the covariates and treatments of $\hat{\mathbb{D}}$. As these are all the required inputs to Algorithm 1, we can compute the optimal balancing weight for this counterfactual dataset $\mathbb{D}$, which we denote as $\hat{\boldsymbol{\alpha}}$.

Notice that the sample average treatments of $\mathbb{D}$ are $\hat{\mathbb{D}}$ should be the same, as they are defined for the same set of units. Therefore the two weighted estimators are approximating the same $\tau_{SATE}$ (or ATE when $N$ increases) and thus

$$\sum_{i \in \mathbb{T}} \alpha_i \mathbb{E}(Y_i(1) \mid \boldsymbol{X}_i) - \sum_{i \in \mathbb{C}} \alpha_i \mathbb{E}(Y_i(0) \mid \boldsymbol{X}_i)$$
$$\approx \sum_{i \in \mathbb{T} \setminus \{1\}} \hat{\alpha}_i \mathbb{E}(Y_i(1) \mid \boldsymbol{X}_i) - \sum_{i \in \mathbb{C}} \hat{\alpha}_i \mathbb{E}(Y_i(0) \mid \boldsymbol{X}_i) - \hat{\alpha}_0 \mathbb{E}(\hat{Y}_1(0) \mid \boldsymbol{X}_1).$$

Therefore we have the following approximation

$$\hat{\alpha}_1 \mathbb{E}(\hat{Y}_1(0) \mid \boldsymbol{X}_1) \approx -\alpha_1 Y_1(1) + \sum_{i \in \mathbb{T} \setminus \{1\}} (\hat{\alpha}_i - \alpha_i) Y_i(1) - \sum_{i \in \mathbb{C}} (\hat{\alpha}_i - \alpha_i) Y_i(0).$$

As we have access to all individual terms on the right, we can compute an approximation of $\mathbb{E}(Y_1(0) \mid \boldsymbol{X}_1)$, using this formula as long as $\hat{\alpha}_0 \neq 0$.[2]

---

[2]Once we have these estimands, policy evaluation can done via plug-in estimations.

To enhance the robustness of this estimation, we can also compute this for units with covariates closed to $\boldsymbol{X}_1$, e.g., using KNNs (Devroye et al., 1994; Li & Tran, 2009), which would give consistent estimations for conditional expectations. Algorithm 4 summarizes this procedure, where Algorithm 3 can be used instead of Algoritm 1 to estimate ITE in a zero-shot fashion.

---

**Algorithm 4** CInA for ITE.

---

1: **Input**: Covariates $\boldsymbol{X}$ and treatments $\boldsymbol{W}$.
2: **Output**: Estimation of $\mathbb{E}(Y_1(1) - Y_1(0) \mid \boldsymbol{X}_1)$.
3: Hyper-parameter: penalty weight $\lambda > 0$.
4: Initialize $\tau = \varnothing$.
5: **for** unit $i$ with $\boldsymbol{X}_i \approx \boldsymbol{X}_1$ **do**
6:     Run Algorithm 1 on $\boldsymbol{X}, \boldsymbol{W}$ to obtain $\boldsymbol{\alpha}$.
7:     Set $\hat{\boldsymbol{W}}$ to be $\boldsymbol{W}$ except $\hat{W}_i = -W_i$.
8:     Run Algorithm 1 on $\boldsymbol{X}, \hat{\boldsymbol{W}}$ to obtain $\hat{\boldsymbol{\alpha}}$.
9:     Let $\hat{\alpha}_i \mathbb{E}(\hat{Y}_i(1 - T_i) \mid \boldsymbol{X}_i) = -\alpha_i Y_i(T_i) + \sum_{j \neq i, T_j = T_i}(\hat{\alpha}_j - \alpha_j)Y_j(T_j) - \sum_{T_j \neq T_i}(\hat{\alpha}_j - \alpha_j)Y_j(T_j)$.
10:     Append $W_i \cdot (\mathbb{E}(\hat{Y}_i(1 - T_i) \mid \boldsymbol{X}_i) - Y_i(T_i))$ to $\tau$ if $\hat{\alpha}_i \neq 0$.
11: **return** Average of $\tau$.

---

# G    DATASET DETAILS

The details of the datasets for simulation A are provided in Section 5.1. We now provide the details of ER-5000 and the real-world datasets. Code for downloading and pre-processing these datasets will be provided upon publication.

**ER-5000.** Each of the ER-5000 datasets is generated following the structural causal model (SCM) framework. The detailed procedure is as follows. First, we sample a random directed acyclic graph (DAG) from the Erdős-Rényi random graph model (Erdős & Rényi, 1960) with edge probability sampled from 0.25 to 0.5. Then, Based on the sampled DAG, we sample the corresponding functional relationships using a linear weight sampler, with random weights sampled from a uniform distribution between 0 and 3. Next, a treatment node and effect node is randomly chosen. For each non-treatment node, we use additive gaussian random noise with standard deviation randomly sampled uniformly between 0.2 and 2. For treatment node, we specify a Bernoulli distribution with logit equal to the functional output of the corresponding node. Finally, we simulate each variable (in $\boldsymbol{X}$, $T$ and $Y$) using the sampled DAG, functional relationships, and noises.

**IHDP and IHDP-resampled.** The Infant Health and Development Program (IHDP) dataset is a semi-dataset complied by Hill (2011). We use the existing versions from Chernozhukov et al. (2022), which are sampled using the outcome model implemented as setting A in (Dorie, 2016). Each dataset comprises of 747 units and 25 covaraites measuring the aspects of children and their mothers. For IHDP, the treatment group (139 out of 747 units) has been made imbalanced by removing a biased subset of the treated population. A total of 1000 datasets are used (following Shi et al. (2019)), where different datasets only differ in terms of outcome values. For IHDP-resampled, 100 datasets are used where the treatments are resampled by setting the propensity score to "True" in the (Dorie, 2016).

**Twins.** Introduced by Louizos et al. (2017), this is a semi-synthetic dataset based on the real data on twin births and twin mortality rates in the US from 1989 to 1991 (Almond et al., 2005). The treatment is "born the heavier twin", which is simulated as a function of the GESTAT10 covariates. Therefore this dataset is confounded. After assigning the treatment for each pair of twins, the dataset is constructed by hiding the other twin. We downloaded the dataset and processed it following Neal et al. (2020).

**LaLonde CPS and PSID.** We also use the datasets from LaLonde (1986), in which the treatment is job training and the outcomes are income and employment status after training. The ground-truth average treatment effect is computed using a randomized study, where we use the observational data to estimate it. The observational data has multiple versions. We use both the PSID-1 and CPS-1 versions for our experiments (Dehejia & Wahba, 1999).

**ACIC.** The data for the 2018 Atlantic Causal Inference Conference competition (ACIC) (Shimoni et al., 2018) comprises of serveral semi-synthetic datasets derived from the linked birth and infant death (LBIDD) data (MacDorman & Atkinson, 1998). The data-generating process is described in (Shimoni et al., 2018). In our experiment, we use datasets containing $1k$ or $10k$ samples.[3] In the experiments in Section 5, a total of 293 datasets (each of size $1k$) were used, where 93 were left out for testing. In Appendix I, we extend this to datasets of size $10k$, where a total of 288 datasets were used and 88 among these were left out for testing. We use datasets with polynomial link function for training and validation. For testing, we use datasets with exponential link functions thus creating a harder task for evaluating our methods.

# H    IMPLEMENTATION DETAILS

Code for our method will be released on GitHub upon publication. Below we describe the architecture, hyper-parameters, training procedures and other details of our method. We also provide the implementation details of the baselines. Finally, we discuss a new data augmentation technique that we observe to be helpful on certain datasets.

## H.1    CINA

**Pre-processing and Padding.** For Algorithm 2, we might encounter multiple datasets with different number of samples. We wish them to share the same transformation from $W, K$ to $V \in \mathbb{R}^{N \times 1}$, where $N$ is the number of units in the corresponding dataset. For this, we adopt similar pre-processing steps as in natural language. We pad all datasets to the same size (i.e., adding dumy units to smaller datasets) and save the masks that indicate these paddings. During back-propagation, we use this mask to make sure that the loss function is only computed using actual units.

**Model Configurations.** We describe the architecture used in Algorithm 2, as the single-dataset version uses the same components aside from parametrizing the values $V$ directly as learnable parameters. An illustration of the forward pass is provided in Figure 2.

For the transformation from covariates $X$ to keys $K$, we implemented two versions: (1) an identical mapping followed by a batch-norm layer $K = \text{bn}(X)$, (2) a projected mapping followed by a batch-norm layer $k_i = \text{bn} \circ \text{relu} \circ \text{linear}(X_i)$. In our first simulation study in Section 5.1, we observe that the projection to be marginally helpful and thus report all the results based on the identical mapping.

For the transformation from $W, K$ to $V$, we first embed $W_i, k_i$ into a 32-dimensional space using one layer of $\text{relu} \circ \text{linear}(\cdot)$. These two 32-dimensional vectors are then concatenated into a 64-dimensional vector following by a batch-norm layer. Denote these 64-dimensional embedding for each unit as $E = [e_1, ..., e_N]^\top$. We encode them into $N \times 1$-dimensional outputs $O$ using a scaled product attention with value, key, query being linear transformations of $E$. Notice that we read off the balancing weights via $V/h(X)W$ and $h(X) \succ 0$. As the optimal weights $\alpha^* \succeq 0$, the values $V$ should have the same sign as $W$ in an element-wise fashion. Therefore to enforce this, we include another multiplier layer to obtain $V$ from the outputs $O$, namely, $V = \text{relu}(OW)$.

**Normalization.** As the optimal balancing weights is in $\mathbb{A} = \{ \mathbf{0} \preceq \boldsymbol{\alpha} \preceq \mathbf{1}, \ \sum_{i \in \mathbb{T}} \alpha_i = \sum_{i \in \mathbb{C}} \alpha_i = 1 \}$, we normalize the read-off balancing weights during inference. In particular, in Algorithm 1 and Algorithm 3, after setting $\boldsymbol{\alpha}^* = \lambda \cdot V/h(X)W$, we project it into $\mathbb{A}$ by taking $\max(\boldsymbol{\alpha}^*, \mathbf{0})$ and normalizing the treated and control group to sum up to 1.

**Hyper-parameters.** For both Algorithm 1 and Algorithm 2, we search for the optimal penalty $\lambda > 0$ from range $[\lambda_{\min}, \lambda_{\max}]$ by exponentially increasing it from $\lambda_{\min}$ to $\lambda_{\max}$. On the same dataset, this range remains the same for both algorithms (and all variations, if applicable). The following table summarizes the values of $\lambda_{\min}$ to $\lambda_{\max}$ for different datasets.

**Training and Evaluations.** For all the experiments, we use a cosine annealing schedule for the learning rate from $l_{\max}$ to $l_{\min}$ during the first half of the training epochs. Then the learning rate is fixed to $l_{\min}$ for the second half of the training epochs. The exact values of $l_{\max}$ and $l_{\min}$ for

---

[3]In datasets with large sample sizes, techniques for efficient transformers (Child et al., 2019; Kitaev et al., 2020; Katharopoulos et al., 2020; Sun et al., 2023) can be applied to accelerate our method.

| Dataset | $\lambda_{\min}$ | $\lambda_{\max}$ |
|---|---|---|
| Simulation A | 1e-6 | 1e-2 |
| Simulation B | 1e-6 | 1e-2 |
| IHDP | 1 | 1000 |
| IHDP-resmapled | 1e-5 | 1000 |
| Twins | 1e-8 | 1e-2 |
| LaLonde CPS | 1e-10 | 5e-6 |
| LaLonde PSID | 1e-10 | 5e-6 |
| ACIC | 1e-6 | 100 |

Table 1: Search range for $\lambda$ in different datasets.

different datasets can be found in the codebase. For Algorithm 1, we train for $20,000$ epochs on all datasets. For Algorithm 2, we train for $4,000$ epochs on all datasets.

For evaluating the results of Algorithm 2, we choose the best hyper-parameters based on the mean absolute error on the validation sets of datasets and report the results on the testing sets of datasets. For evaluating the results of Algorithm 1, if the setting contains multiple datasets (Simulation A, Simulation B, IHDP-resampled, ACIC), we choose the best hyper-parameters based on the mean absolute error on the validation sets of datasets and report the results on the testing sets of datasets. Note that even though IHDP contains multiple datasets, they all share the same sets of covariates and treatments. Therefore we treat it the same as settings with one dataset for Algorithm 1. On these datasets (IHDP, Twins, LaLonde CPS, LaLonde PSID), we choose the best hyper-parameters based on the reported results.

## H.2 BASELINES

**IPW and Self-Normalized IPW.** For both IPW and self-normalized IPW, we first standardized the covariates $X$. Then we fit a random forest classifier on the data to predict propensity scores. The depth of the random forest classifier is chosen in the same way as the hyper-parameter $\lambda$ is chosen in CInA, which we described above.

**DML.** For DML, we use the implementation of Battocchi et al. (2019). In particular, we consider three models: `LinearDML`, `CausalForestDML`, `KernelDML`. Similar as above, when a validation set of datasets is present, we report the results based on the best of these three models in terms of validation MAE. Otherwise we report based on the best performance on the reported dataset. However, in simulation A, we only use `LinearDML` as the outcome model is linear.

**SVM.** For this baseline, we first standardized the covariates $X$. Then we solve the dual SVM problem in Eq. (2), where the kernel is defined using $\phi$ given in Eq. (9) on the standardized data. We use the support vector classifier (Pedregosa et al., 2011) with a precomputed kernel. The maximum number of iterations is capped with a hard limit of $50,000$. The reported results are based on $\lambda$ choosen in the same way as CInA described above.

## H.3 DATASET AUGMENTATION

In our experiments in Section 5.1 and certain datasets in Section 5.3 using the multi-dataset version of CInA, we implemented a new type of data augmentation. As we observe that the network can learn how to balance on a set of datasets using very few training steps, we propose to reshuffle amongst different datasets in every epoch. This essentially creates a "new" set of datasets by combining units from different datasets. Intuitively, this augments the number of covariate balancing problems that the model has to learn to solve without actually needing to acquire more data. However, we note that this technique is only applied if different datasets from the same experiment share the same causal graph. If different datasets contain very different causal structures such as **ER-5000** in Section 5.2 and **ACIC** in Section 5.3, this shuffling is not used as it would create covariate balancing problem that does not aid learning. The main intuition is that if we reshuffle units among these datasets, units in a reshuffled dataset could follow different causal graphs, which means there is potentially no underlying causal structure that can explain the data.

# I  ADDITIONAL EMPIRICAL RESULTS

## I.1  COMPARISON TO DRAGONNET AND RIESZNET

| Method | Simulation-A | ER-5000 | IHDP |
|---|---|---|---|
| Naive | 0.172 ± 0.03 | 50.27 ± 5.97 | 0.259 ± 0.01 |
| IPW | 0.304 ± 0.03 | 27.42 ± 3.19 | 0.766 ± 0.02 |
| Self-normalized IPW | 0.158 ± 0.03 | 49.99 ± 5.88 | 0.141 ± 0.00 |
| DML | 0.094 ± 0.01 | 11.13 ± 3.17 | 0.585 ± 0.03 |
| DragonNet | 0.386 ± 0.01 | 11.21 ± 3.17 | 0.146 ± 0.01 |
| RieszNet | **0.045 ± 0.01** | 12.90 ± 4.54 | **0.110 ± 0.01** |
| SVM | 0.015 ± 0.00 | 11.09 ± 3.13 | 1.202 ± 0.05 |
| Ours | 0.126 ± 0.02 | N/A | **0.114 ± 0.01** |
| Ours (ZS) | 0.147 ± 0.01 | 11.50 ± 1.85 | N/A |
| Ours (ZS-S) | N/A | **2.66 ± 0.33** | N/A |
| Mean | N/A | 17.88 ± 1.83 | N/A |

Table 2: ATE MAE comparison of different methods on the "Simulation-A", "ER-5000", and "IHDP" datasets.

In this section, we further compare two additional baselines, DragonNet (Shi et al., 2019) and RieszNet (Chernozhukov et al., 2022), both of which were considered strong neural estimation methods for per-dataset causal inference. Results for **IHDP** dataset were directly cited from (Shi et al., 2019; Chernozhukov et al., 2022), following their best performing models. Furthermore, we also compare to **Simulation-A-Multi+OOD+diff_size**, and **ER-5000**, both are the most general synthetic settings in Section 5. On **Simulation-A-Multi+OOD+diff_size**, *CINA (ZS)* outperforms *DragonNet*, while *RieszNet* outperforms both *DragonNet* and *CINA (ZS)* method. On both **ER-5000** and **IHDP**, *CINA (ZS)* is on par with or outperforms *DragonNet* and *RieszNet*, while *CINA (ZS-S)* massively outperforms the other methods on **ER-5000**.

## I.2  LARGER SCALE EXPERIMENTS ON 10K ACIC 2018, WITH CROSS-DATASET GENERALIZATION

| Method | ATE MAE | Inference time on new data (s) | Pretraining time (s) |
|---|---|---|---|
| Naive | 13.07 ± 8.25 | 0.005 | N/A |
| IPW | 10.29 ± 5.94 | 48.927 | N/A |
| Self-normalized IPW | 10.30 ± 5.90 | 49.322 | N/A |
| DML | 8.572 ± 8.96 | 7391.743 | N/A |
| RieszNet | 69.39 ± 31.9 | 8157.498 | N/A |
| Ours (ZS) | **1.460 ± 0.48** | 78.503 | 1800 |
| Ours (ZS-S) | **1.361 ± 0.42** | 77.546 | 1800 |
| Ours (ZS-ER) | 1.718 ± 0.74 | 78.085 | 1800 |
| Ours (ZS-S-ER) | 1.702 ± 0.74 | 77.947 | 1800 |

Table 3: Comparison of different methods on the 10k ACIC 2018 dataset.

To demonstrate the performance of our method on larger version of **ACIC 2018**, we produce additional experiment using the 10k-size datasets of ACIC (Shimoni et al., 2018), which is a commonly used scale considered in the literature (Shi et al., 2019; Mahajan et al., 2022). Note that instead of only selecting a subset of datasets in **ACIC 2018** as in (Shi et al., 2019; Mahajan et al., 2022), we make use of all datasets of size 10k generated by (Shimoni et al., 2018) that has polynomial link functions as training datasets, and all datasets of size 10k with exponential link functions as test datasets.

In this setting, we also compare two new variants of our method, *CINA (ZS-ER)* and *CINA (ZS-S-ER)*, that are fully trained on a larger-scale, 200-dimensional ER-5000 dataset Section 5.2 under both unsupervised and supervised settings, respectively. After pre-training, *CINA (ZS-ER)* and

*CINA (ZS-S-ER)* are applied directly to all ACIC 2018 test sets. This will help us to demonstrate whether the model can show generalization ability across datasets. All CINA-related methods are trained for a fixed time budget (1800 seconds), which is significantly shorter than the full training time of DML and RieszNet. As shown in Table 2, both *CINA (ZS)* and *CINA (ZS-S)* significantly outperforms all baselines. The *CINA (ZS-ER)* and *CINA (ZS-S-ER)* methods give marginally worse performance than *CINA (ZS)* and *CINA (ZS-S)*, but still out-performs the other baselines by a clear margin.

