# OpenReview forum: "Towards Causal Foundation Model: on Duality between Causal Inference and Attention"
_ICLR.cc/2024/Conference — Submitted to ICLR 2024_

### Official Review · Reviewer_FDY4 · 2023-10-31

**Soundness:** 2 fair
**Presentation:** 1 poor
**Contribution:** 2 fair
**Rating:** 5
**Confidence:** 5

**Summary:**

Inspired by the success of foundation models on text and image, this paper proposes a foundation model for causal inference. The model is able to perform zero-shot causal inference.

**Strengths:**

1. Causal effect estimation is an important task and proposing a foundation model to conduct this task is interesting.
2. Estimating the causal effect in a zero-shot manner is interesting,

**Weaknesses:**

1. The writing of the paper needs to be improved. The abstract and introduction is very misleading. It is only after reading the main framework section that the reader will realize the proposed method is essentially a spacial kind of large foundation model which is designed to perform causal inference and the main contribution of the paper is to not make foundation models causal, i.e., make them rely on the causal relations rather than the spurious correlations.

2. The paper needs to provide an intuition of why this model works. What kind of information is learned by this model that enables zero-shot causal inference. Though it is relatively intuitive when it comes to text and image based foundation models, it needs further intuition and justification for causal effect estimation.

3. As a followup to the previous point, the  experiments are not comprehensive. ALso more information on the model implementation and experimental set is needed.

**Questions:**

Please refer to weaknesses

---

> ### Author Response · Authors · 2023-11-18
> **Reply to Reviewer FDY4**
>
> We appreciate that the reviewer found our zero-shot method to be interesting, and our studied problem to be important. We would like to address the reviewer’s concerns below:
>
> > **“The writing of the paper needs to be improved. The abstract and introduction is very misleading. It is only after reading the main framework section that the reader will realize the proposed method is essentially a special kind of large foundation model which is designed to perform causal inference and the main contribution of the paper is to not make foundation models causal, i.e., make them rely on the causal relations rather than the spurious correlations.”**
>
> We thank the reviewer for this suggestion. We have modified the abstract, introduction and the related work section to clarify that we are proposing a method for building a foundation model for causal inference tasks, instead of modifying the existing foundation models.
>
> In particular, we significantly revised the second paragraph of the introduction to provide a context of existing foundation models’ ability on inferring causal quantities (see also the related work section for a detailed discussion). This then motivates and transitions into our proposal on how to design a method specialized for causal inference tasks, which is summarized in detail in the contributions. In addition, we also revised the abstract and the contribution paragraph to reflect this.
>
> In summary, the introduction (and abstract) now provides a summary of the importance of foundation models, the existing foundation models’ capability on causal inference tasks and why we are motivated to build causal foundation models, the difficulties, and our proposal. We hope this clarifies the confusion on the main contribution of this paper.
>
> > **“The paper needs to provide an intuition of why this model works. What kind of information is learned by this model that enables zero-shot causal inference. Though it is relatively intuitive when it comes to text and image based foundation models, it needs further intuition and justification for causal effect estimation.”**
>
> In essence, we use attention to learn to produce optimal balancing weights that **debias** any given dataset. Then, treatment effect estimation can be estimated easily. We expect this to generalize well, since our theory suggests that attentions (under the correct loss function and assumptions) are **exact** optimal balancing solvers.
>
> In particular, our proposed method applies the attention mechanism to different units and tries to balance among the units. The optimal balancing weights are learned using the proposed loss in Eq. (7). In the zero-shot setting, we amortize the learning procedure and encode the procedure that produces the optimal balancing weight using a neural network (its parameters are denoted by $\phi$). Intuitively, once this is learned using multiple datasets, the model learns how to solve the optimal balancing problem. This is similar to learning how to optimize `[1]` or learning how to solve combinatorial optimization problems `[2]`, where the model acquires the ability to solve a new task (i.e., balance in a new dataset in our context). We discussed this briefly after introducing Algorithm 2 and 3, but we have edited it to further reflect and clarify the intuitions.
>
> > **“As a followup to the previous point, the experiments are not comprehensive. Also more information on the model implementation and experimental set is needed.”**
>
> Regarding comprehensiveness: as causal inference tasks necessitate counterfactual information for validation and evaluation, experiments and benchmarks are often done on synthetic datasets or carefully curated real-world datasets. We included almost all common real-world causal inference datasets used by various previous works (c.f., benchmarks in `[3,4]`) and also created different simulations of varying difficulties. In addition, in revision (Appendix I.1), we included comparisons to more baselines: Dragonnet `[5]`, and RieszNet `[6]`, both considered as state-of-the-art neural estimation methods. We have also added larger scale ACIC experiments in Appendix I.2. We would be happy to add additional experiments if there are specific ones that we are missing.
>
> Regarding the model implementation and the experimental setup: apart from model architecture, hyperparameters, evaluations, and etc, which are provided in Appendix G and H, we have added the details of the data-generating process of simulation B to Appendix G. The data generating process of simulation A provided in Section 5.1 of the main text. We will also release the source code once this work becomes public.

---

> > ### Author Response · Authors · 2023-11-18
> > **References**
> >
> > [1] Bengio, Yoshua, Andrea Lodi, and Antoine Prouvost. "Machine learning for combinatorial optimization: a methodological tour d’horizon." European Journal of Operational Research 290.2 (2021): 405-421.
> >
> > [2] Kool, Wouter, Herke Van Hoof, and Max Welling. "Attention, learn to solve routing problems!." arXiv preprint arXiv:1803.08475 (2018).
> >
> > [3] Brady Neal, Chin-Wei Huang, and Sunand Raghupathi. Realcause: Realistic causal inference benchmarking. arXiv preprint arXiv:2011.15007, 2020.
> >
> > [4] Mahajan, Divyat, et al. "Empirical analysis of model selection for heterogenous causal effect estimation." arXiv preprint arXiv:2211.01939 (2022).
> >
> > [5] Shi, Claudia, David Blei, and Victor Veitch. "Adapting neural networks for the estimation of treatment effects." Advances in neural information processing systems 32 (2019).
> >
> > [6] Chernozhukov, Victor, et al. "Riesznet and forestriesz: Automatic debiased machine learning with neural nets and random forests." International Conference on Machine Learning. PMLR, 2022.

---

> > > ### Comment · Reviewer_FDY4 · 2023-11-22
> > > **Thank you for your response!**
> > >
> > > I'd like to thank the authors for their comprehensive response. I have fully read all the rebuttal arguments and other reviews. Although some of the issues were answered I still believe some of the main issues that were raised by reviewers still stand. Most importantly, I strongly believe that the proposed model is far from a foundation model based on causal relations and is an application of foundation models to a very specific task of causality (i.e., causal effect estimation). At any rate, I very much so appreciate the authors' effort to respond to my concerns.

---

> ### Author Response · Authors · 2023-11-22
> **Opportunity to engage in discussion and answer further questions**
>
> Dear reviewers, we thank you again for taking the time to review our paper. We are grateful to all the feedback and comments that help improve our work.
>
> Following the suggestions, we have updated the manuscript (summarized in our general response) and tried to address the raised questions in our previous responses. With the end of the discussion period drawing near, we would love to see if our responses address the concerns and whether you are willing to raise the score. We would also be happy to engage in discussions and offer further clarifications if there are additional questions.
>
> Thank you again for your thoughtful considerations.

---

> ### Author Response · Authors · 2023-11-23
> **New results and clarifications to address your concerns**
>
> Thank you for acknowledging our rebuttal!
>
> - We would like to give you an update that all the remaining concerns from other reviewers have now been addressed. As a key criteria requested by reviewer **deJS** (who originally raised the concern regarding whether the model is foundational), we have conducted the **requested experiment train on ER-5000, evaluate on larger scale 10k ACIC**, in both supervise and unsupervised settings.  Full results can be found in **Appendix I.2**, with the key numbers (with ER pretrained model boldfaced) summarized as below:
>
> |Method|ATE MAE|Inference time on new data (s)| Pretraining time (s)|
> | ---------------------------------------------------- | ------------------------- | ------------------------- |------------------|
> |Naive|13.07 ± 8.25|0.005| N/A|
> |IPW|10.29 ± 5.94|48.927| N/A|
> |Self-normalized IPW|10.30 ± 5.90|49.322| N/A|
> |DML|8.572 ± 8.96|7391.743| N/A|
> |RieszNet| 69.39 ± 31.9|8157.498| N/A|
> |Ours (ZS)|1.460 ± 0.48|78.503|1800|
> |Ours (ZS-S)|1.361 ± 0.42|77.546|1800|
> |**Ours (ZS-ER**)|1.718 ± 0.74|78.085|1800|
> |**Ours (ZS-S-ER**)|1.702 ± 0.74|77.947|1800|
>
>  As shown by the results The CINA (ZS-ER) (trained on ER, unsupervised, evaluate on ACIC) and CINA (ZS-S-ER) (trained on ER, supervised, evaluated on ACIC) still **out-performs the other baselines, including DML and RieszNet, by a clear margin**.  This demonstrate that our method is able to generalize across dataset, and hence by the criteria from reviewer **deJS**, our contribution should qualify as a first step towards building causal foundation models.
>
>
> - We agree that building a casual foundation model still requires many efforts. This is why we emphasize in our paper that this work as **_only_** a first step. Importantly, there are a number of existing work in the foundation model literature [1, 2, 3, 4, 5] that **shares the same formulation** (i.e., applied on **single task** but focus on generalization **across datasets**). We hereby argue that this is a commonly adopted approach and should not serve as a legitimate counterpoint to our contribution.
>
> - However, we understand that there could be different subjective understandings. Therefore as a further clarification to this main point, we change the title to **“Causal Inference with Attention: On Duality between Optimal Balancing and Attention”**, so that the contribution is clear. In addition, we would be happy to address other remaining specific issues within our scope if there are any.
>
> We hope these address your concerns and thank you again for your time and valuable feedbacks!
>
> **References**
>
> [1] Zhang, Han, et al. "Towards Foundation Models for Learning on Tabular Data." arXiv preprint arXiv:2310.07338 (2023).
>
> [2] Rasul, Kashif, et al. "Lag-Llama: Towards Foundation Models for Time Series Forecasting." arXiv preprint arXiv:2310.08278 (2023).
>
> [3] Das, Abhimanyu, et al. "A decoder-only foundation model for time-series forecasting." arXiv preprint arXiv:2310.10688 (2023).
>
> [4] Garza, Azul, and Max Mergenthaler-Canseco. "TimeGPT-1." arXiv preprint arXiv:2310.03589 (2023).
>
> [5] Galkin, Mikhail, et al. "Towards Foundation Models for Knowledge Graph Reasoning." arXiv preprint arXiv:2310.04562 (2023).

---

### Official Review · Reviewer_YMpy · 2023-10-31

**Soundness:** 3 good
**Presentation:** 3 good
**Contribution:** 3 good
**Rating:** 6
**Confidence:** 2

**Summary:**

This paper delves into the intricate relationship between causal inference and attention mechanisms in transformer-like models, understanding these models through a causal lens. To address this, they introduce a novel method called Causal Inference with Attention (CInA), which utilizes self-supervised causal learning, enabling zero-shot causal inference on unseen tasks. The method's foundation lies in the primal-dual connection between optimal covariate balancing and self-attention, enabling inference through a transformer-type architecture's final layer. Empirical experiments validate CInA's effectiveness, showing its capability to generalize across out-of-distribution and real-world datasets, often matching or outperforming traditional causal inference methodologies.

**Strengths:**

1. This paper is well-written and well-organized.

2. The introduction of the Causal Inference with Attention (CInA) method is interesting and theoretically sound.

**Weaknesses:**

I am not very familiar with this topic and I may change my score after further discussions with authors and other reviewers.

1. Did the paper exaggerate its findings regarding "Casual foundation models"? From what I gathered, the study introduced a method to understand the attention mechanism in transformer-type networks through a causal lens. However, they only tested their approach on relatively small datasets. This makes it challenging to view this work as a blueprint for developing foundation models, and its efficacy for such models remains unproven. While the related work section delves into causal reasoning for Large Language Models (LLMs), the connection between the proposed method and LLMs isn't clearly articulated.

2. This paper purports to be an initial step towards creating causal foundation models. But how does it compare to paper [1]? And could you elucidate the link between the causal transformer and the methods they proposed?

[1] Causal Transformer for Estimating Counterfactual Outcomes.

3.  The methods you've presented can be challenging to understand, particularly for those not well-versed in the subject. It would be helpful to include an introductory section that provides background information. After that, you can highlight your contributions.

**Questions:**

Please refer to the section above.

---

> ### Author Response · Authors · 2023-11-18
> **Reply to Reviewer YMpy (Part 1)**
>
> Thank you for acknowledging our proposed method, and for finding this paper to be well written! We would like to address your questions below:
>
> > **“Did the paper exaggerate its findings regarding "Casual foundation models"? From what I gathered, the study introduced a method to understand the attention mechanism in transformer-type networks through a causal lens. However, they only tested their approach on relatively small datasets. This makes it challenging to view this work as a blueprint for developing foundation models, and its efficacy for such models remains unproven.”**
>
> We have further clarified the implication of foundation models in our revision (Appendix A). Our response can be summarized as below:
> - In the context of causal foundation models, the word **foundation** emphasizes more on the ability for a single model to generalize to OOD datasets. In the ER-5000 experiment, we trained a model on 5k different datasets, which generalizes well to 1k different OOD datasets generated by different causal structures/mechanisms. We argue that this qualifies as evidence that our method has the potential towards building causal foundation models.
> - The scale of the largest dataset (~10k) considered in our paper is on par with typical causal inference papers. For instance, the largest datasets considered by DragonNet `[2]` is at the scale of 10k, while RieszNet `[3]` considers ~3k. These works were considered as state-of-the-art neural estimators for the task we are considering.
> - Compared to some existing foundation models for tabular and time series datasets, the scale of our experiment is not much smaller on the order of magnitude. For example, `[4]` operates on datasets on the scale of 1k ~ 50k; similar holds for `[5]`. Note that in terms of context length of the trained model, ours (1024) is larger than both `[4,5]` where 512 is used.
> - Going forward, we consider extending the scope of empirical efforts by gathering and curating larger scale real-world/semi-synthetic data as an important future step. This was briefly discussed in the conclusion. However, much work remains to be done to build a large (public) dataset for obtaining a fully pretrained model, and thus we consider it out of scope for this work.
>
> > **“While the related work section delves into causal reasoning for Large Language Models (LLMs), the connection between the proposed method and LLMs isn't clearly articulated.”**
>
> Thank you for this comment. The related work section on “Causal Reasoning with Large Language Models” aims to convey that LLMs have difficulties with problems involving high numerical precisions (such as ATE estimation), and that we are taking a different approach towards building causally-aware foundation models (i.e., constructing from fundamentals instead of prompting/modifying existing LLMs). Aside from this, there is no further connection between this work and LLMs except that both use the attention mechanisms. We apologize for the confusion caused by this. In the revised manuscript, we have highlighted this discussion in a separate paragraph in the corresponding related work section.
>
> > **“This paper purports to be an initial step towards creating causal foundation models. But how does it compare to paper [1]? And could you elucidate the link between the causal transformer and the methods they proposed?”**
>
> We thank the reviewer for this pointer and have added this discussion to the related work discussion of the revised manuscript (Appendix B).
>
> The method proposed by `[1]` is different from our work on a few aspects.  First, `[1]` trains their model on a **per-dataset** basis; and whenever a new dataset comes in, their model needs to be re-trained. Thus, this still falls into the realm of traditional, non-foundational causal models. While in our proposed framework towards building causal foundation models, we aim to train a single model that can perform zero-shot inference to a range of different contexts/datasets.
>
> Second, although both works make use of transformer structures, in their method, the transformer attends to the **time dimension for the same unit**, and produces a conditional estimand of the outcome of a single unit at the next step. While in our method, the attention is applied across different units to produce an estimand of the average effect across different units.

---

> ### Author Response · Authors · 2023-11-18
> **Reply to Reviewer YMpy (Part 2)**
>
> > **“The methods you've presented can be challenging to understand, particularly for those not well-versed in the subject. It would be helpful to include an introductory section that provides background information. After that, you can highlight your contributions.”**
>
> We thank the reviewer for this suggestion, we have modified the introduction to include a short description of the problem under study (highlighted in contributions), as well as the related work section to include pointers to background information in Section 3.1 (highlighted in “Causal Inference via Optimal Balancing”).
>
> The problem formulation of estimating average treatment effect is summarized in the first paragraph of Section 3.1. The method we proposed is following the literature on optimal covariance balancing, which produces an estimand for the treatment effect based on re-weighting different units in a dataset (see our response to the above question of how attention is used to produce balancing weights). We have provided a discussion on optimal covariance balancing in Section 2.
>
> -----------
> **References**
>
> [1] Melnychuk, Valentyn, Dennis Frauen, and Stefan Feuerriegel. "Causal transformer for estimating counterfactual outcomes." International Conference on Machine Learning. PMLR, 2022.
>
> [2] Shi, Claudia, David Blei, and Victor Veitch. "Adapting neural networks for the estimation of treatment effects." Advances in neural information processing systems 32 (2019).
>
> [3] Chernozhukov, Victor, et al. "Riesznet and forestriesz: Automatic debiased machine learning with neural nets and random forests." International Conference on Machine Learning. PMLR, 2022.
>
> [4] Zhang, Han, et al. "Towards Foundation Models for Learning on Tabular Data." arXiv preprint arXiv:2310.07338 (2023).
>
> [5] Das, Abhimanyu, et al. "A decoder-only foundation model for time-series forecasting." arXiv preprint arXiv:2310.10688 (2023).

---

### Official Review · Reviewer_BTeh · 2023-11-02

**Soundness:** 4 excellent
**Presentation:** 4 excellent
**Contribution:** 4 excellent
**Rating:** 10
**Confidence:** 3

**Summary:**

In this paper, the authors present a step towards creating a self-supervised learning paradigm for causal foundation models, which they call Causal Inference with Attention (CInA), a transformer-based model that uses covariate balancing as self-supervised tasks and is trained on multiple unlabeled observational datasets for treatment effect estimation, backed by the primal-dual connection between covariate balancing and regularized self-attention. Their study is backed by both synthetic and real-world datasets to show good zero-shot generalisation performance with moderate causal mechanism shifts considering the metric of mean average error (MAE) between the true Average Treatment Effect (ATE) and the predicted ATE in both the single causal graph setting and when the data comes from multiple different causal graphs.

**Strengths:**

- Causally aware foundation models that are realisable for complex tasks such as inferring ATE on complex datasets like ER-5000, Twins dataset, and IHDP is an important step in deep learning, in general.
- The contribution of this paper showing zero-shot causal inference on unseen data is a big strength, and its MAE is shown to be lower than even per-dataset causal inference approaches.
- The authors prove the duality between covariate balancing and self-attention, showing that under mild regularity conditions, the optimal parameters of the self-attention layers provide optimal covariate balancing weighs and ATE.

**Weaknesses:**

- Just a minor suggestion since this is also a part of the paper already-- it could potentially be useful to state the causal reasoning task explicitly, such as estimating ATE with greater generalizability.
- The source code is not available, but the authors say they will release it upon publication.

**Questions:**

- I couldn't understand the connection between different causal graphs and covariate balancing-- the authors say they shuffle units from different datasets using training-- how is the differentiation between how different the causal structures are, performed? And why does this not help with increasing the number of covariate balancing problems without acquiring more data?

**Details Of Ethics Concerns:**

No ethics concerns.

---

> ### Author Response · Authors · 2023-11-18
> **Reply to Reviewer BTeh**
>
> Thank you for the encouraging comments! We appreciate that the reviewer found our studied problem to be important and our proposed method and its zero-shot ability to be significant. We will open source the code upon publication. In addition, we would to address your comments as follows.
>
> > **“Just a minor suggestion since this is also a part of the paper already-- it could potentially be useful to state the causal reasoning task explicitly, such as estimating ATE with greater generalizability.”**
>
> Thank you for this suggestion, we have modified the abstract and introduction to state our causal reason task of treatment effect estimations (highlighted in abstract and contributions). In addition, we included a discussion on the definition and implication of (causal) foundation models in Appendix A to further clarify our causal inference tasks.
>
> > **“I couldn't understand the connection between different causal graphs and covariate balancing-- the authors say they shuffle units from different datasets using training-- how is the differentiation between how different the causal structures are, performed? And why does this not help with increasing the number of covariate balancing problems without acquiring more data?”**
>
> In the dataset augmentation described in Appendix H.3, we added another paragraph to clarify this point. This reshuffling trick is only applied to a set of datasets that share the **same causal graph** in the data-generation process. For example, we do this for simulation A and IHDP-resampled (this shares the same causal graph because different datasets share the same set of covariates and the treatments/outcomes are sampled conditioned on these covariates).
>
> For a set of datasets where each contains units generated by a causal graph that varies between datasets, e.g. ER-5000 and ACIC, this is disabled. The main intuition is that if we reshuffle units among these datasets, units in a reshuffled dataset could follow different causal graphs, which means there is potentially no underlying causal structure that can explain the data.

---

### Official Review · Reviewer_deJS · 2023-11-06

**Soundness:** 1 poor
**Presentation:** 2 fair
**Contribution:** 2 fair
**Rating:** 3
**Confidence:** 4

**Summary:**

The paper claims three contributions aimed towards the problem of zero-shot causal inference with transformer models, as a step towards "causally-aware foundation models":
1. A theoretical equivalence between optimal covariate balancing under the assumption of back-door admissibility of covariates (ignorability) and self-attention with a particular loss function.
2. A algorithm implementing covariate balancing using transformers.
3. Experiments illustrating improvements in model performance and runtime.

**Strengths:**

- Originality: The idea of linking self-attention, SVM-like optimization, and an estimator for the classic back-door adjustment (a.k.a. optimal covariate balancing) is a novel one to my knowledge.
- Significance: The work introduces a method that can perform competitively with prior sample-reweighting methods such as DML in small (n<=1024) datasets. It also introduces a "zero-shot" method which trains on multiple datasets and is evaluated on a single new dataset; this method significantly outperforms prior methods in performance and speed due to the fact that re-training is not required, when trained and evaluated on datasets similar to the new dataset (e.g., when trained on ER-5000 datasets, it outperforms DML on ER-5000, and when trained on ACIC datasets, it outperforms DML on other ACIC datasets).
- Clarity: The notation and claimed contributions in the paper are generally clear, and the paper is generally well-written, aside for key points mentioned in the weaknesses section.

**Weaknesses:**

Overall, I believe the framing of the work as a step towards causal foundation models is misleading, as the method introduced is limited to estimating the ATE under the classic ignorability assumption. The theoretical novelty of the work does not seem strong, as Theorem 1 is a direct implication of results from prior work cited in the paper (Tarr & Imai, 2021; Kallus, 2020; Nguyen et al., 2022). Finally, the strength of the empirical results is tempered by the facts that there is no comparison in performance to various neural ATE estimation methods, and that it is not possible to rule out overfitting as an explanation for outperformance of zero-shot (supervised) learning (see point 7).

1. The title "Towards Causal Foundation Model" seems to be misleading, along with the claim "we take a first step towards building causal foundation models, focusing on causal inference tasks". The paper does use self-attention, a key element of most 'foundation models' in the language or vision sense. However, the proposed method is limited to estimation of the ATE under the assumptions of ignorability and sample non-interference; indeed, it relies on the structure of the problem to produce a weighted average of observed outcomes. According to (Bommasani et al., 2021), "A foundation model is any model that is trained on broad data (generally using self-supervision at scale) that can be adapted (e.g., fine-tuned) to a wide range of downstream tasks; current examples include BERT [Devlin et al. 2019], GPT-3 [Brown et al. 2020], and CLIP [Radford et al. 2021]." Clearly, this does not hold for a method that can only estimate the ATE under a specific set of assumptions, independent of any efficiency or performance improvements.
2. The assumption of sample non-interference, along with the fact that the self-attention is applied on the units of a dataset, seems to remove any connection with the "Causal Inference with Large Language Models" section, as one main reason attention is useful in language models is the fact that it handles sequences well. In theory, arbitrarily re-ordering a dataset should not affect the output of Algorithm 1, due to the non-interference assumption. Re-ordering the words in a document inputted to an LLM *should*, of course, affect the output of an LLM. Therefore, it is difficult to see any connection between self-attention as used in this work and as used in LLMs - or any connection between this work and LLMs, beyond the fact that it is known that LLMs have difficulty solving the problem of ATE estimation from data and assumptions.
3. Section 3.1 appears not to contain novel theoretical results; the results showing optimal covariate balancing is equivalent to the dual SVM problem appear to be repeated from (Tarr & Imai, 2021; Kallus, 2020). Section 3.2 appears to rely on (Nguyen et al., 2022) for its result showing the dual SVM problem may be reformulated as self-attention. Therefore, the novelty of Theorem 1 does not seem to be particularly strong, as it is a direct implication of prior work's results.
4. Algorithm 1 seems to be limited to estimating the ATE - that is, the method does *not* extend to counterfactual estimation, as described in Appendix D. The **sample** average treatment effect, or SATE, (not to be confused with the average treatment effect, or ATE) is not identifiable under the assumptions of consistency, non-interference, and ignorability ($Y_i(0), Y_i(1) \perp T_i | \mathbf X_i$). As a simple demonstrative example, let $n = 1$, and consider the single sample $(X = 1, Y = 1)$. It is possible for $Y_1(0)$ (and therefore the sample average treatment effect) to take the value of 0 or 1, and in the absence of other assumptions, we cannot infer its value. Adding more samples will not help us infer its value. This is also known as the fundamental problem of causal inference (Holland, 1986), and the causal hierarchy theorem (Bareinboim et al., 2022) shows that it is in general not possible to identify counterfactual queries from observational data alone, which is what is specified in the problem formulation.
5. There is no comparison to methods such as (Shi et al., 2019), cited in the paper, which is a neural estimation method for the average treatment effect. There is also no comparison to (Xia et al., 2021; Xia et al., 2022; https://causalai.net/r80.pdf, https://causalai.net/r87.pdf), a neural identification and estimation technique for arbitrary interventional and counterfactual queries.
6. It is unclear how the method will scale to datasets of sample size n > 1024, as evaluation time scales quadratically with n. For example, ACIC contains 100k samples, but the experiments subsample datasets of size <=1000, as described in Appendix E. The comparison with ACIC seems unfair, because other methods could learn from the 100k-sample dataset in at most linearly scaling time, while the proposed method's runtime would scale quadratically.
6. It is unclear whether or not the method in the experiments is overfitting to the class of datasets used, since the training and test datasets are drawn from the same dataset generator or kind of real dataset. If the goal is truly to illustrate "foundation model"-like properties, experiments showing cross-dataset generalization would be required. Not enough details are given on the generation of the ER-5000 dataset for me to ascertain that overfitting on the dataset class is not occuring. For example, if the self-attention model is trained on ER-5000, it should be tested on datasets from simulation study A and real datasets, such as IHDP, Twins, or ACIC; if it is trained on real datasets of one variety (e.g., ACIC), it should be tested on datasets of another (e.g., IHDP). Other methods do not have the benefit of training on multiple datasets, so this would be a fair comparison betwen the zero-shot (supervised) learning method and other methods.

References:
1. Bareinboim, Elias, et al. "On Pearl’s hierarchy and the foundations of causal inference." Probabilistic and causal inference: the works of judea pearl. 2022. 507-556.
2. Holland, Paul W. "Statistics and causal inference." Journal of the American statistical Association 81.396 (1986): 945-960.

**Questions:**

1. How can the current work be viewed as a "causal foundation model", or related, given that it is targeted towards solving the particular task of ATE estimation under a particular set of assumptions?
2. Given that the self-attention used in the paper lacks the classic sequence-learning benefits of self-attention in LLMs due to the assumption of sample non-interference, what is the connection between this work and LLMs?
3. Are there novel theoretical results in this work beyond direct implications from (Tarr & Imai, 2021; Kallus, 2020; Nguyen et al., 2022)?
4. Are there any assumptions that I missed in the paper which would allow one to identify or infer any bound on the sample ATE when $n = 1$ and we have the single sample $(X = 1, Y = 1)$?
5. I am curious to know if training on multiple datasets to make inferences on a new one may have induced overfitting on the class of datasets used. For example, what is the performance of the ZS method when trained on ER-5000 (supervised or unsupervised) and evaluated on ACIC?
6. I understand that the method proposed in the paper likely cannot scale well to dataset of ~100k samples, such as ACIC 2018. However, it would be informative to apply DML to each full dataset of 100k and compare this performance to the performance of CInA when given an allowance of the same runtime.
7. Shouldn't the estimator in Section 3.1 minimize the expected absolute (or squared) conditional bias, rather than the absolute value of the expected conditional bias? That is, shouldn't there be an absolute value inside the expectation, or shouldn't the difference be squared? If not, then one can have mean-zero error with arbitrarily high variance, which does not seem to describe a good estimator.

---

> ### Author Response · Authors · 2023-11-18
> **Reply to Reviewer deJS (Part 1)**
>
> Thank you for the detailed comments! We appreciate that the reviewer found our idea of using self-attention and SVM-like optimization for covariate balancing novel, and our algorithm to be significant. We would like to address the reviewer’s comments below:
>
> > **“How can the current work be viewed as a "causal foundation model", or related, given that it is targeted towards solving the particular task of ATE estimation under a particular set of assumptions?”**
>
>
> The **task-universality** of foundation models does not necessarily require adaptability across different **ML formulations** (e.g., prediction, imputation, ATE, counterfactuals); it can also refer to adaptability across different **contexts** of a given task. This viewpoint is shared by many recent studies on foundation models beyond LLMs `[1,2,3,4,5]`, where they all focus on one single type of task but evaluate in-context generalization across datasets. In our case, we focus on treatment effect estimation tasks and evaluate our model in unseen datasets/contexts. We have clarified this implication of foundation models in our revision (Appendix A).
>
> Ignorability and non-interference are classic canonical assumptions that ensure ATE and CATE identifiability.  In practice, our method performs well even with certain levels of violations, as shown in ER-5000 and real-world datasets. Evidence from literature `[6,7]` indicates that optimal balancing can still be a good objective when certain assumptions are relaxed, e.g., when there is latent confounding. We acknowledge the importance of these extensions; however, they fall into orthogonal directions to this work.
>
> As clarified throughout the paper, our work is intended as a stepping stone towards causal foundation models, not a final solution. We have emphasized this in our title, introduction, and the revised abstract. We acknowledge the importance of solving causal problems beyond ATE under relaxed assumptions. Nevertheless, these require much more development that is beyond the scope of this paper and would be best addressed in separate follow-up works.
>
> > **“What is the connection between this work and LLMs?” / “lacks the classic sequence-learning benefits of self-attention in LLMs”**
>
> We apologize for the confusion caused by the related work section on “Causal Reasoning with Large Language Models”. This section was intended to highlight the limitations of LLMs in quantitative causal tasks such as ATE estimation; we did not imply that there is any deep connection between this work and LLMs, since attention was used for very different purposes. We further clarified this by adding the discussion in the corresponding section.
>
> > **“Are there novel theoretical results in this work beyond direct implications from (Tarr & Imai, 2021; Kallus, 2020; Nguyen et al., 2022)?”**
>
> The novelty of our result is clarified below. We only borrow `(Nguyen et al., 2022)` for the existence of feature maps in Eq. (6); however the SVM optimization problem derived from our formulation is different from `(Nguyen et al., 2022)`. Moreover, our theoretical result focuses on gradient-based optimization in Eq. (7) whereas `(Nguyen et al., 2022)` focusing on deriving new attention architectures. Finally, we would like to highlight that our result gives rise to a novel and important algorithmic implication that attention can be exactly mapped as a treatment effect solver. This enables a theory-grounded method for zero-shot causal inference. To the best of our knowledge, this finding is new.
>
> > **“Algorithm 1 seems to be limited to estimating the ATE - that is, the method does not extend to counterfactual estimation, as described in Appendix D.” / “Are there any assumptions that I missed in the paper that would allow one to identify or infer any bound on the sample ATE when we have the single sample?”**
>
> We apologize for the confusion and have further clarified and revised this part accordingly (now Appendix F). Similar to certain previous works `[8,9]`, the term “counterfactuals” was used interchangeably with "individual treatment effects (ITEs)" in our original submission. In the latest revision, we have clarified that the algorithm in Appendix D (now F) is intended to serve as an alternative method for estimating ITE/CATE. Under the same set of assumptions (e.g., ignorability), ITE/CATE is identifiable. Note that our estimation for ATEs (or ITEs) requires access to a dataset with multiple units, not just a single sample. We did not suggest that a sample ATE can be identified from a single sample. We have updated our manuscript (Appendix F) to make this clearer. In Appendix F, we also added the pseudo-code for ITE estimation, which is based on Algorithm 1.

---

> ### Author Response · Authors · 2023-11-18
> **Reply to Reviewer deJS (Part 2)**
>
> > **“There is no comparison to methods such as (Shi et al., 2019) ... There is also no comparison to (Xia et al., 2021; Xia et al., 2022) ...”**
>
> Thank you for this suggestion. We added a comparison to two popular neural estimation methods: Dragonnet `(Shi et al., 2019)` and RieszNet `[10]`, which can be found in Appendix I.1. We summarize the results below for simulation A (Multi+OOD+diffsize), ER-5000, and IHDP datasets.
>
> |Method|Simulation-A |ER-5000|IHDP|
> | ----------------------------------------- | ------------------------- | ------------------------- | ------------------------- |
> |DragonNet|0.386 ± 0.01|11.21 ± 3.17|0.146 ± 0.01|
> |RieszNet|**0.045 ± 0.01**|12.90 ± 4.54|**0.110 ± 0.01**|
> |Ours|0.126 ± 0.02|N/A|**0.114 ± 0.01**|
> |Ours (ZS)|0.147 ± 0.01|11.50 ± 1.85|N/A|
> |Ours (ZS-S)|N/A|**2.66 ± 0.33**|N/A|
>
> We observe that our zero-shot method in general is still on-par or better than Dragonnet and RieszNet. This indicates that the proposed method can reach current per-dataset state-of-the-art go-to methods, even under zero-shot inference settings
>
> We did not compare to `(Xia et al., 2021 & 2022)` as their method is centered around G-consistent NCMs, which require access to the true graph of the underlying dataset. However, we included a full discussion of these neural estimation methods in Appendix B.
>
> > **“It is unclear how the method will scale to datasets of sample size n > 1024…” / “I understand that the method proposed in the paper likely cannot scale well to dataset of ~100k samples, such as ACIC 2018. However, it would be informative to apply DML to each full dataset of 100k and compare this performance to the performance of CInA when given an allowance of the same runtime.”**
>
>
> Thank you for this comment. We would like to address this using the following points:
> - The original ACIC dataset `[11]` includes pre-generated datasets of varying sizes (1k, 2.5k, 5k, 10k, 25k, 50k), each with different data generation mechanisms. We used datasets of size 1k without any further down-sampling in the experiment section. We clarified this in Appendix G.
> - During revision, we conducted additional experiments on larger ACIC datasets of size 10k, a common choice of scale considered in the previous benchmarking (`Shi et al., 2019` and `[12]`) This was added to Appendix I.2. We attached the results below. Our method still significantly outperformed other baselines.
> - Regarding the current model scale, we would like to highlight that our context length (1024) is already larger than some existing foundational models for tabular and time series datasets, such as `[1,3]` which used 512.
> - Our method, with its transformer-like structure, can benefit from efficient transformer techniques `[13,14,15,16]` to handle datasets of larger scale. We added this point to the discussion section.
>
> |Method|ATE MAE|Elapsed Time (s)|
> | ---------------------------------------------------- | ------------------------- | ------------------------- |
> |Naive|13.07 ± 8.25|0.005|
> |IPW|10.29 ± 5.94|48.927|
> |Self-normalized IPW|10.30 ± 5.90|49.322|
> |DML|8.572 ± 8.96|7391.743|
> |RieszNet| 69.39 ± 31.9|8157.498|
> |Ours (ZS)|**1.460 ± 0.48**|78.503|
> |Ours (ZS-S)|**1.361 ± 0.42**|77.546|
>
>
> > **“It is unclear whether or not the method in the experiments is overfitting to the class of datasets used, since the training and test datasets are drawn from the same dataset generator or kind of real dataset.”**
>
> Our experimental settings are carefully curated to control for and prevent overfitting. We would like to clarify that in many settings, we use different data generators for the training and testing datasets. For example,
> - In variation 2 of simulation A (_multi-mechanism ood setting_), we use a different mechanism to generate each dataset, and ensure that they have non-overlapping support for the coefficient parameter of $P(T|X)$ between training and testing.
> - For the ER-5000 experiment, each dataset was generated using a different causal graph, with minimal similarities between training and test datasets.
> - Lastly, for the ACIC 2018 datasets, we used polynomial link functions for training and validation, while test datasets were all generated by exponential link functions.
>
> >**“what is the performance of the ZS method when trained on ER-5000 (supervised or unsupervised) and evaluated on ACIC?”**
>
> While we acknowledge the importance of transferring to distinct real-world datasets, one major challenge lies the current scale of available datasets. The scale of the synthetic ER-5000 dataset is not enough for the model to learn how to balance in a very different dataset such as ACIC. Going forward, we view it as an important future step to extend the scope of empirical efforts by gathering and curating large-scale real-world/-synthetic data. We have discussed this in the conclusion. However, much work remains to be done to build a large (public) dataset for obtaining a fully pretrained model, and thus we consider it out of scope for this work.

---

> ### Author Response · Authors · 2023-11-18
> **Reply to Reviewer deJS (Part 3)**
>
> >**“Shouldn't the estimator in Section 3.1 minimize the expected absolute (or squared) conditional bias, rather than the absolute value of the expected conditional bias? That is, shouldn't there be an absolute value inside the expectation, or shouldn't the difference be squared? If not, then one can have mean-zero error with arbitrarily high variance, which does not seem to describe a good estimator.”**
>
>
> We used the absolute value of the expected conditional bias because it provides a closed-form upper bound in Eq. (1), which simplifies the optimization problem. If the outcome has bounded variance, Theorem 2 in `[7]` ensures the estimator's variance will be upper bounded by the square of the expected conditional bias (up to a multiplication by constant and addition by $O(1/N)$). Therefore, it can't have low bias with arbitrarily high variance.
>
> --------------------------------
> **References**
>
> [1] Zhang, Han, et al. "Towards Foundation Models for Learning on Tabular Data." arXiv preprint arXiv:2310.07338 (2023).
>
> [2] Rasul, Kashif, et al. "Lag-Llama: Towards Foundation Models for Time Series Forecasting." arXiv preprint arXiv:2310.08278 (2023).
>
> [3] Das, Abhimanyu, et al. "A decoder-only foundation model for time-series forecasting." arXiv preprint arXiv:2310.10688 (2023).
>
> [4] Garza, Azul, and Max Mergenthaler-Canseco. "TimeGPT-1." arXiv preprint arXiv:2310.03589 (2023).
>
> [5] Galkin, Mikhail, et al. "Towards Foundation Models for Knowledge Graph Reasoning." arXiv preprint arXiv:2310.04562 (2023).
>
> [6] Kallus, Nathan. "Deepmatch: Balancing deep covariate representations for causal inference using adversarial training." International Conference on Machine Learning. PMLR, 2020.
>
> [7] Bennett, Andrew, and Nathan Kallus. "Policy evaluation with latent confounders via optimal balance." Advances in neural information processing systems 32 (2019).
>
> [8] Shalit, Uri, Fredrik D. Johansson, and David Sontag. "Estimating individual treatment effect: generalization bounds and algorithms." International conference on machine learning. PMLR, 2017.
>
> [9] Assaad, Serge, et al. "Counterfactual representation learning with balancing weights." International Conference on Artificial Intelligence and Statistics. PMLR, 2021.
>
> [10] Chernozhukov, Victor, et al. "Riesznet and forestriesz: Automatic debiased machine learning with neural nets and random forests." International Conference on Machine Learning. PMLR, 2022.
>
> [11] Shimoni, Yishai, et al. "Benchmarking framework for performance-evaluation of causal inference analysis." arXiv preprint arXiv:1802.05046 (2018).
>
> [12] Mahajan, Divyat, et al. "Empirical analysis of model selection for heterogenous causal effect estimation." arXiv preprint arXiv:2211.01939 (2022).
>
> [13] Child, Rewon, et al. "Generating long sequences with sparse transformers." arXiv preprint arXiv:1904.10509 (2019).
>
> [14] Kitaev, Nikita, Łukasz Kaiser, and Anselm Levskaya. "Reformer: The efficient transformer." arXiv preprint arXiv:2001.04451 (2020).
>
> [15] Katharopoulos, Angelos, et al. "Transformers are rnns: Fast autoregressive transformers with linear attention." International conference on machine learning. PMLR, 2020.
>
> [16] Sun, Yutao, et al. "Retentive network: A successor to transformer for large language models." arXiv preprint arXiv:2307.08621 (2023).

---

> ### Author Response · Authors · 2023-11-22
> **Opportunity for discussing or answering further questions**
>
> Dear reviewer, we thank you again for your time and the valuable feedback, which helps improve our paper.
>
> Following the comments, we have updated the manuscript (summarized in our general response) and tried to address the questions you raised in our previous responses.  As the discussion end date approaches, we would love to see if our responses adequately answer the questions and whether you are willing to change your recommendation. We are happy to engage in discussions and offer further clarifications if there are additional questions.
>
> Once again, thank you for the thoughtful considerations.

---

> ### Comment · Reviewer_deJS · 2023-11-22
> **I have read the authors' response.**
>
> I thank the authors for their in-depth and thoughtful response. Unfortunately, due to time constraints, I will be unable to engage in significant discussion with the authors. I will share a more in-depth response when I have more availability.
>
> Three brief comments:
> 1. **Calling the current work a step towards causal foundation models seems too strong to describe the contributions in the paper.** It seems that supervised CInA cannot generalize between datasets of different types (e.g. ER-5000 to ACIC), which would be a requirement for a foundation model. For example, an LLM foundation model would be expected to generalize between books and newspaper articles, despite significant differences in writing styles. It seems this type of capability is out of the scope of the current work. Given that the zero-shot learning contribution seems to be the only connection between CInA and foundation models and that the aforementioned type of zero-shot learning is out of scope, I do not see the connection between the current work and foundation models (let me know if I'm mistaken - I discussed in my review how the use of attention is not a connection between CInA and foundation models, and how LLMs seem to be unrelated to CInA, and I do not see other potential connections between the current work and foundation models). This would certainly be an interesting direction to explore in the future, as the authors mentioned in the rebuttal, but calling the current work a step towards causal foundation models seems too strong to describe the contributions in the paper.
> 2. **I am not convinced that overfitting is not occurring in the supervised CInA experiments.** The details provided do not convince me that the training and testing distributions are significantly different for the supervised CInA experiments. The query I requested (train on ER-5000, evaluate on ACIC) would be sufficient to show that overfitting is not occuring, but no empirical result of equivalent strength was provided.
> 3. **Does CInA still perform competitively with DML when given the same runtime budget?** CInA's methodology is to my knowledge novel, and it performs competitively with DML when given the same number of samples, though it doesn't clearly outperform prior methods in the unsupervised setting, and it doesn't outperform the SVM method it is based on in simulated settings. I'd be curious to know a) how its runtime (training included, for the unsupervised case) scales on larger datasets and b) if it still performs competitively with DML if both methods are provided with the same runtime budget.
>
> Overall, I believe there is non-trivial novelty in the work, and I find the work very interesting. However, my belief remains that the work's significance (particularly, supervised results + connection to foundation models) is overstated given the empirical results provided in the paper and rebuttal.

---

> > ### Author Response · Authors · 2023-11-23
> > **Response to further comments -- new results for requested experiments**
> >
> > Thank you again for the additional comments! We understand the time constraints and would like to provide succinct responses to the additional comments:
> >
> > > **Calling the current work a step towards causal foundation models seems too strong to describe the contributions in the paper...CInA cannot generalize between datasets of different types (ER-5000 to ACIC),**
> >
> > > **I am not convinced that overfitting is not occurring in the supervised CInA experiments....The query I requested (train on ER-5000, evaluate on ACIC) would be sufficient to show that overfitting is not occuring, but no empirical result of equivalent strength was provided.**
> >
> > To address both of your concerns, we have conducted the **requested experiment train on ER-5000, evaluate on larger scale 10k ACIC**, in both supervise and unsupervised settings.  Full results can be found in **Appendix I.2**, with the key numbers (with ER pretrained model boldfaced) summarized as below:
> >
> > |Method|ATE MAE|Inference time on new data (s)| Pretraining time (s)|
> > | ---------------------------------------------------- | ------------------------- | ------------------------- |------------------|
> > |Naive|13.07 ± 8.25|0.005| N/A|
> > |IPW|10.29 ± 5.94|48.927| N/A|
> > |Self-normalized IPW|10.30 ± 5.90|49.322| N/A|
> > |DML|8.572 ± 8.96|7391.743| N/A|
> > |RieszNet| 69.39 ± 31.9|8157.498| N/A|
> > |Ours (ZS)|1.460 ± 0.48|78.503|1800|
> > |Ours (ZS-S)|1.361 ± 0.42|77.546|1800|
> > |**Ours (ZS-ER**)|1.718 ± 0.74|78.085|1800|
> > |**Ours (ZS-S-ER**)|1.702 ± 0.74|77.947|1800|
> >
> > As shown by the results The CINA (ZS-ER) (trained on ER, unsupervised, evaluate on ACIC) and CINA (ZS-S-ER) (trained on ER, supervised, evaluated on ACIC) still **out-performs the other baselines, including DML and RieszNet, by a clear margin**.
> >
> >
> > Meanwhile, we would like to emphasize againthat this work is _only_ a first step. However, we understand that there can be different opinions / understandings. Therefore as a further clarification in response to this point, we have changed the title to **“Causal Inference with Attention: On Duality between Optimal Balancing and Attention”**, so that the contribution is clear.
> >
> > > **Does CInA still perform competitively with DML when given the same runtime budget?**
> >
> > - Short answer: **yes**. We kindly point to **Appendix I.2** for the additional experiment comparing CInA to prior methods in _larger-scale_ ACIC datasets, where the zero-shot _CInA (ZS)_ (including its ER-trained version as requested) outperforms all baselines significantly. Regarding the **training time**, we have added the pre-training time of CInA to the appendix for the ACIC experiment. Please note that in all our results for CInA models (supervised or unsupervised, ER pretrained or not) are obtained with a fixed **pre-training time budged of 1800 seconds** (30 mins). This is significantly lower than RieszNet (8157s) and DML (7391s with EconML), while being able to significantly outperform both baselines, including the unsupervised setting.
> >
> >
> > We greatly appreciate that the reviewer finds merits with our paper, and sincerely hope that these responses provided have fully addressed your queries and concerns.

---

### Author Response · Authors · 2023-11-18
**Summary of Revisions**

We thank the reviewers again for their encouraging reviews. In particular, we appreciate that reviewers found our proposal to be novel (reviewer deJS, BTeh, YMpy), the zero-shot performance to be significant (reviewer deJS, BTeh), and the problem of causal foundation model under study to be important (reviewer BTeh, FDY4).

We also appreciate the valuable suggestions which have helped improve our manuscript. Below, we summarize the key revisions.
* we carried out significant **discussions/clarifications of causal foundation models**
    - to _clarify the definition of (causal) foundation models_ and why we consider _this work as a step towards building causal foundation models_, we added Appendix A which describes the implication/definition of (causal)  foundation models.
    - to clarify _our contributions in the bigger picture of causal foundation models_, we revised both abstract and introduction to state the specific causal inference task, that is treatment effect estimation with greater generalizability.
    - to clarify _the relationship between this work and LLMs_, we significantly revised the second paragraph of the introduction and the corresponding related work section.
* we carried out **additional benchmarking experiments**:
    - we added two _additional neural estimation baselines_ (Dragonnet `[1]` and Riesznet `[2]`) in our comparison. The results can be found in Appendix I.1, where our method shows on-par and sometimes better performance while being much faster.
    - we increased the _scale_ of experiments on ACIC to consider datasets with size 10k, a common choice of scale considered in the previous benchmarking (`[1]`, `[3]`). This result can be found in Appendix I.2, where our methods show improved performance both in terms of accuracy and speed.
* we added an additional literature review of **related neural estimation methods** in Appendix B

We hope these can address the concerns raised by the reviewers and further clarify the value of this work.

---------------
**References**

[1] Shi, Claudia, David Blei, and Victor Veitch. "Adapting neural networks for the estimation of treatment effects." Advances in neural information processing systems 32 (2019).

[2] Chernozhukov, Victor, et al. "Riesznet and forestriesz: Automatic debiased machine learning with neural nets and random forests." International Conference on Machine Learning. PMLR, 2022.

[3] Mahajan, Divyat, et al. "Empirical analysis of model selection for heterogenous causal effect estimation." arXiv preprint arXiv:2211.01939 (2022).

---

### Author Response · Authors · 2023-11-23
**Summary of 2nd Round Revisions**

We thank the reviewers again for responding to our rebuttals. Below, we summarize the key revisions that addresses all the remaining concerns from both **reviewer deJS** and **reviewer FDY4**.

- As specifically requested by **reviewer deJS**: as a quantitative criteria to show that our work qualifies as a first step towards building causal foundation models, we have conducted the **requested experiment train on ER-5000, evaluate on larger scale 10k ACIC**, in both supervise and unsupervised settings.  Full results can be found in **Appendix I.2**, with the key numbers (with ER pretrained model boldfaced) summarized as below:

|Method|ATE MAE|Inference time on new data (s)| Pretraining time (s)|
| ---------------------------------------------------- | ------------------------- | ------------------------- |------------------|
|Naive|13.07 ± 8.25|0.005| N/A|
|IPW|10.29 ± 5.94|48.927| N/A|
|Self-normalized IPW|10.30 ± 5.90|49.322| N/A|
|DML|8.572 ± 8.96|7391.743| N/A|
|RieszNet| 69.39 ± 31.9|8157.498| N/A|
|Ours (ZS)|1.460 ± 0.48|78.503|1800|
|Ours (ZS-S)|1.361 ± 0.42|77.546|1800|
|**Ours (ZS-ER**)|1.718 ± 0.74|78.085|1800|
|**Ours (ZS-S-ER**)|1.702 ± 0.74|77.947|1800|

As shown by the results The CINA (ZS-ER) (trained on ER, unsupervised, evaluate on ACIC) and CINA (ZS-S-ER) (trained on ER, supervised, evaluated on ACIC) still **out-performs the other baselines, including DML and RieszNet, by a clear margin**.  This should address the concerns raised by both **reviewer deJS** and **reviewer FDY4**.


- As requested by **reviewer deJS**, we have added the pre-training time of CInA to Appendix I.2, where CInA models (supervised or unsupervised, ER pretrained or not) are trained with a fixed **pre-training time budged of 1800 seconds** (30 mins), which is significantly lower than RieszNet (8157s) and DML (7391s with EconML), while being able to significantly outperform both baselines.


- To further clarify our contribution, we have changed the title to **“Causal Inference with Attention: On Duality between Optimal Balancing and Attention”**.


We hope these can address the concerns raised by the reviewers and further clarify the value of this work.

---

### Meta-Review · Area_Chair_j6zN · 2023-12-04

**Metareview:**

In this paper, the authors present a step towards creating a self-supervised learning paradigm for causal foundation models, which they call Causal Inference with Attention (CInA), a transformer-based model which utilizes self-supervised causal learning, enabling zero-shot causal inference on unseen tasks.

However, the issues raised by the reviews are critical, for instance, the theoretical novelty (deJS, YMpy), and the possible misleading and clarity issues (deJS, FDY4).  Authors need to address those raised concerns. The decision is reject.

**Justification For Why Not Higher Score:**

The issues of theoretical novelty and the possible misleading and clarity issues that are raised by the reviewers.

**Justification For Why Not Lower Score:**

N/A

---

### Decision · Program_Chairs · 2024-01-16

Reject